# Sparse Logistic Regression Learns All Discrete Pairwise Graphical Models

**Shanshan Wu, Sujay Sanghavi, Alexandros G. Dimakis**
Department of Electrical and Computer Engineering
University of Texas at Austin
shanshan@utexas.edu, sanghavi@mail.utexas.edu, dimakis@austin.utexas.edu

## Abstract

We characterize the effectiveness of a classical algorithm for recovering the Markov graph of a general discrete pairwise graphical model from i.i.d. samples. The algorithm is (appropriately regularized) maximum conditional log-likelihood, which involves solving a convex program for each node; for Ising models this is $\ell_1$-constrained logistic regression, while for more general alphabets an $\ell_{2,1}$ group-norm constraint needs to be used. We show that this algorithm can recover any arbitrary discrete pairwise graphical model, and also characterize its sample complexity as a function of model width, alphabet size, edge parameter accuracy, and the number of variables. We show that along every one of these axes, it matches or improves on all existing results and algorithms for this problem. Our analysis applies a sharp generalization error bound for logistic regression when the weight vector has an $\ell_1$ (or $\ell_{2,1}$) constraint and the sample vector has an $\ell_\infty$ (or $\ell_{2,\infty}$) constraint. We also show that the proposed convex programs can be efficiently solved in $\tilde{O}(n^2)$ running time (where $n$ is the number of variables) under the same statistical guarantees. We provide experimental results to support our analysis.

## 1 Introduction

Undirected graphical models provide a framework for modeling high dimensional distributions with dependent variables and have many applications including in computer vision (Choi et al., 2010), bio-informatics (Marbach et al., 2012), and sociology (Eagle et al., 2009). In this paper we characterize the effectiveness of a natural, and already popular, algorithm for the *structure learning* problem. Structure learning is the task of finding the dependency graph of a Markov random field (MRF) given i.i.d. samples; typically one is also interested in finding estimates for the edge weights as well. We consider the structure learning problem in general (non-binary) discrete pairwise graphical models. These are MRFs where the variables take values in a discrete alphabet, but all interactions are pairwise. This includes the Ising model as a special case (which corresponds to a binary alphabet).

The natural and popular algorithm we consider is (appropriately regularized) maximum conditional log-likelihood for finding the neighborhood set of any given node. For the Ising model, this becomes $\ell_1$-constrained logistic regression; more generally for non-binary graphical models the regularizer becomes an $\ell_{2,1}$ norm. We show that this algorithm can recover all discrete pairwise graphical models, and characterize its sample complexity as a function of the parameters of interest: model width, alphabet size, edge parameter accuracy, and the number of variables. We match or improve dependence on each of these parameters, over all existing results for the general alphabet case when no additional assumptions are made on the model (see Table 1). For the specific case of Ising models, some recent work has better dependence on some parameters (see Table 2 in Appendix A).

We now describe the related work, and then outline our contributions.

| Paper | Assumptions | Sample complexity ($N$) |
|---|---|---|
| Greedy algorithm (Hamilton et al., 2017) | 1. Alphabet size $k \geq 2$<br>2. Model width $\leq \lambda$<br>3. Degree $\leq d$<br>4. Minimum edge weight $\geq \eta > 0$<br>5. Probability of success $\geq 1 - \rho$ | $O\big(\exp\big(\frac{k^{O(d)}\exp(O(d^2\lambda))}{\eta^{O(1)}}\big)\ln\big(\frac{nk}{\rho}\big)\big)$ |
| Sparsitron (Klivans and Meka, 2017) | 1. Alphabet size $k \geq 2$<br>2. Model width $\leq \lambda$ | $O\big(\frac{\lambda^2 k^5 \exp(14\lambda)}{\eta^4}\ln\big(\frac{nk}{\rho\eta}\big)\big)$ |
| $\ell_{2,1}$-constrained logistic regression (**this paper**) | 3. Minimum edge weight $\geq \eta > 0$<br>4. Probability of success $\geq 1 - \rho$ | $O\big(\frac{\lambda^2 k^4 \exp(14\lambda)}{\eta^4}\ln\big(\frac{nk}{\rho}\big)\big)$ |

Table 1: Sample complexity comparison for different graph recovery algorithms. The pairwise graphical model has alphabet size $k$. For $k = 2$ (i.e., Ising models), our algorithm reduces to the $\ell_1$-constrained logistic regression (see Table 2 in Appendix A for related work on learning Ising models). Our sample complexity has a better dependency on the alphabet size ($\tilde{O}(k^4)$ versus $\tilde{O}(k^5)$) than that in (Klivans and Meka, 2017)[2].

**Related Work**

In a classic paper, Ravikumar et al. (2010) considered the structure learning problem for Ising models. They showed that $\ell_1$-regularized logistic regression provably recovers the correct dependency graph with a very small number of samples by solving a convex program for each variable. This algorithm was later generalized to multi-class logistic regression with $\ell_{2,1}$ group sparse regularization, for learning MRFs with higher-order interactions and non-binary variables (Jalali et al., 2011). A well-known limitation of (Ravikumar et al., 2010; Jalali et al., 2011) is that their theoretical guarantees only work for a restricted class of models. Specifically, they require that the underlying learned model satisfies technical *incoherence* assumptions, that are difficult to validate or check.

A large amount of recent work has since proposed various algorithms to obtain provable learning results for general graphical models without requiring the incoherence assumptions. We now describe the (most related part of the extensive) related work, followed by our results and comparisons (see Table 1). For a discrete pairwise graphical model, let $n$ be the number of variables and $k$ be the alphabet size; define the model width $\lambda$ as the maximum neighborhood weight (see Definition 1 and 2 for the precise definition). For structure learning algorithms, a popular approach is to focus on the sub-problem of finding the neighborhood of a single node. Once this is correctly learned, the overall graph structure is a simple union bound. Indeed all the papers we now discuss are of this type. As shown in Table 1, Hamilton et al. (2017) proposed a greedy algorithm to learn pairwise (and higher-order) MRFs with general alphabet. Their algorithm generalizes the approach of Bresler (2015) to learning Ising models. The sample complexity in (Hamilton et al., 2017) grows logarithmically in $n$, but *doubly* exponentially in the width $\lambda$ (only *single* exponential is necessary (Santhanam and Wainwright, 2012)). Klivans and Meka (2017) provided a different algorithmic and theoretical approach by setting this up as an online learning problem and leveraging results from the Hedge algorithm therein. Their algorithm Sparsitron achieves single-exponential dependence on $\lambda$.

**Our Contributions**

- Our main result: We show that a classical algorithm $\ell_{2,1}$-constrained[3] logistic regression can recover the edge weights of a discrete pairwise graphical model from i.i.d. samples (see Theorem 2). For the special case of Ising models (see Theorem 1), this reduces to an $\ell_1$-constrained logistic regression. For the general setting with non-binary alphabet, since each edge has a group of parameters, it is natural to use an $\ell_{2,1}$ group sparse constraint to enforce sparsity at the level of

groups. We make no incoherence assumption on the graphical models. As shown in Table 1, our sample complexity scales as $\tilde{O}(k^4)$, which improves[4] the previous best result with $\tilde{O}(k^5)$ dependency[5]. Our analysis applies a sharp generalization error bound for logistic regression when the weight vector has an $\ell_{2,1}$ (or $\ell_1$) constraint and the sample vector has an $\ell_{2,\infty}$ (or $\ell_\infty$) constraint (see Lemma 8 and 11 in Appendix E). Our key insight is that a generalization bound can be used to control the squared distance between the predicted and true logistic functions (see Lemma 1 and 2 in Appendix B), which then implies an $\ell_\infty$ norm bound between the weight vectors (see Lemma 5 and 6 in Appendix B).

- We show that the proposed algorithms can run in $\tilde{O}(n^2)$ time without affecting the statistical guarantees (see Section 2.3). Note that $\tilde{O}(n^2)$ is an efficient runtime for graph recovery over $n$ nodes. Previous algorithms in (Hamilton et al., 2017; Klivans and Meka, 2017) also require $\tilde{O}(n^2)$ runtime for structure learning of pairwise graphical models.

- We construct examples that violate the incoherence condition proposed in (Ravikumar et al., 2010) (see Figure 1). We then run $\ell_1$-constrained logistic regression and show that it can recover the graph structure as long as given enough samples. This verifies our analysis and shows that our conditions for graph recovery are weaker than those in (Ravikumar et al., 2010).

- We empirically compare the proposed algorithm with the Sparsitron algorithm in (Klivans and Meka, 2017) over different alphabet sizes, and show that our algorithm needs fewer samples for graph recovery (see Figure 2).

**Notation.** We use $[n]$ to denote the set $\{1, 2, \cdots, n\}$. For a vector $x \in \mathbb{R}^n$, we use $x_i$ or $x(i)$ to denote its $i$-th coordinate. The $\ell_p$ norm of a vector is defined as $\|x\|_p = (\sum_i |x_i|^p)^{1/p}$. We use $x_{-i} \in \mathbb{R}^{n-1}$ to denote the vector after deleting the $i$-th coordinate. For a matrix $A \in \mathbb{R}^{n \times k}$, we use $A_{ij}$ or $A(i, j)$ to denote its $(i, j)$-th entry. We use $A(i, :) \in \mathbb{R}^k$ and $A(:, j) \in \mathbb{R}^n$ to the denote the $i$-th row vector and the $j$-th column vector. The $\ell_{p,q}$ norm of a matrix $A \in \mathbb{R}^{n \times k}$ is defined as $\|A\|_{p,q} = \|[\|A(1, :)\|_p, ..., \|A(n, :)\|_p]\|_q$. We define $\|A\|_\infty = \max_{ij} |A(i, j)|$. We use $\langle \cdot, \cdot \rangle$ to represent the dot product between two vectors $\langle x, y \rangle = \sum_i x_i y_i$ or two matrices $\langle A, B \rangle = \sum_{ij} A(i, j) B(i, j)$.

## 2 Main results

We start with the special case of binary variables (i.e., Ising models), and then move to the general case with non-binary variables.

### 2.1 Learning Ising models

We first give a definition of an Ising model distribution.

**Definition 1.** *Let $A \in \mathbb{R}^{n \times n}$ be a symmetric weight matrix with $A_{ii} = 0$ for $i \in [n]$. Let $\theta \in \mathbb{R}^n$ be a mean-field vector. The $n$-variable Ising model is a distribution $\mathcal{D}(A, \theta)$ on $\{-1, 1\}^n$ that satisfies*

$$\mathbb{P}_{Z \sim \mathcal{D}(A, \theta)}[Z = z] \propto \exp \left( \sum_{1 \le i < j \le n} A_{ij} z_i z_j + \sum_{i \in [n]} \theta_i z_i \right). \tag{1}$$

*The dependency graph of $\mathcal{D}(A, \theta)$ is an undirected graph $G = (V, E)$, with vertices $V = [n]$ and edges $E = \{(i, j) : A_{ij} \ne 0\}$. Define the width of $\mathcal{D}(A, \theta)$ as*

$$\lambda(A, \theta) = \max_{i \in [n]} \left( \sum_{j \in [n]} |A_{ij}| + |\theta_i| \right). \tag{2}$$

*Let $\eta(A, \theta)$ be the minimum edge weight, i.e., $\eta(A, \theta) = \min_{(i,j) \in E} |A_{ij}|$.*

One property of an Ising model distribution is that the conditional distribution of any variable given the rest variables follows a logistic function. Let $\sigma(z) = 1/(1 + e^{-z})$ be the sigmoid function.

**Fact 1.** *Let $Z \sim \mathcal{D}(A, \theta)$ and $Z \in \{-1, 1\}^n$. For any $i \in [n]$, the conditional probability of the $i$-th variable $Z_i \in \{-1, 1\}$ given the states of all other variables $Z_{-i} \in \{-1, 1\}^{n-1}$ is*

$$\mathbb{P}[Z_i = 1 | Z_{-i} = x] = \frac{\exp(\sum_{j \neq i} A_{ij} x_j + \theta_i)}{\exp(\sum_{j \neq i} A_{ij} x_j + \theta_i) + \exp(-\sum_{j \neq i} A_{ij} x_j - \theta_i)} = \sigma(\langle w, x' \rangle), \quad (3)$$

*where $x' = [x, 1] \in \{-1, 1\}^n$, and $w = 2[A_{i1}, \cdots, A_{i(i-1)}, A_{i(i+1)}, \cdots, A_{in}, \theta_i] \in \mathbb{R}^n$. Moreover, $w$ satisfies $\|w\|_1 \leq 2\lambda(A, \theta)$, where $\lambda(A, \theta)$ is the model width defined in Definition 1.*

Following Fact 1, the natural approach to estimating the edge weights $A_{ij}$ is to solve a logistic regression problem for each variable. For ease of notation, let us focus on the $n$-th variable (the algorithm directly applies to the rest variables). Given $N$ i.i.d. samples $\{z^1, \cdots, z^N\}$, where $z^i \in \{-1, 1\}^n$ from an Ising model $\mathcal{D}(A, \theta)$, we first transform the samples into $\{(x^i, y^i)\}_{i=1}^N$, where $x^i = [z_1^i, \cdots, z_{n-1}^i, 1] \in \{-1, 1\}^n$ and $y^i = z_n^i \in \{-1, 1\}$. By Fact 1, we know that $\mathbb{P}[y^i = 1 | x^i = x] = \sigma(\langle w^*, x \rangle)$ where $w^* = 2[A_{n1}, \cdots, A_{n(n-1)}, \theta_n] \in \mathbb{R}^n$ satisfies $\|w^*\|_1 \leq 2\lambda(A, \theta)$. Suppose that $\lambda(A, \theta) \leq \lambda$, we are then interested in recovering $w^*$ by solving the following $\ell_1$-constrained logistic regression problem

$$\hat{w} \in \arg\min_{w \in \mathbb{R}^n} \frac{1}{N} \sum_{i=1}^N \ell(y^i \langle w, x^i \rangle) \qquad \text{s.t. } \|w\|_1 \leq 2\lambda, \qquad (4)$$

where $\ell : \mathbb{R} \to \mathbb{R}$ is the loss function

$$\ell(y^i \langle w, x^i \rangle) = \ln(1 + e^{-y^i \langle w, x^i \rangle}) = \begin{cases} -\ln \sigma(\langle w, x^i \rangle), & \text{if } y^i = 1 \\ -\ln(1 - \sigma(\langle w, x^i \rangle)), & \text{if } y^i = -1 \end{cases} \qquad (5)$$

Eq. (5) is essentially the negative log-likelihood of observing $y^i$ given $x^i$ at the current $w$.

Let $\hat{w}$ be a minimizer of (4). It is worth noting that in the high-dimensional regime ($N < n$), $\hat{w}$ may not be unique. In this case, we will show that *any* one of them would work. After solving the convex problem in (4), the edge weight is estimated as $\hat{A}_{nj} = \hat{w}_j/2$.

The pseudocode of the above algorithm is given in Algorithm 1. Solving the $\ell_1$-constrained logistic regression problem will give us an estimator of the true edge weight. We then form the graph by keeping the edge that has estimated weight larger than $\eta/2$ (in absolute value).

---

**Algorithm 1:** Learning an Ising model via $\ell_1$-constrained logistic regression

---

**Input:** $N$ i.i.d. samples $\{z^1, \cdots, z^N\}$, where $z^m \in \{-1, 1\}^n$ for $m \in [N]$; an upper bound on $\lambda(A, \theta) \leq \lambda$; a lower bound on $\eta(A, \theta) \geq \eta > 0$.
**Output:** $\hat{A} \in \mathbb{R}^{n \times n}$, and an undirected graph $\hat{G}$ on $n$ nodes.

1 **for** $i \leftarrow 1$ **to** $n$ **do**
2 $\quad \forall m \in [N], x^m \leftarrow [z_{-i}^m, 1], y^m \leftarrow z_i^m$
3 $\quad \hat{w} \leftarrow \arg\min_{w \in \mathbb{R}^n} \frac{1}{N} \sum_{m=1}^N \ln(1 + e^{-y^m \langle w, x^m \rangle})$ s.t. $\|w\|_1 \leq 2\lambda$
4 $\quad \forall j \in [n], \hat{A}_{ij} \leftarrow \hat{w}_{\tilde{j}}/2$, where $\tilde{j} = j$ if $j < i$ and $\tilde{j} = j - 1$ if $j > i$
5 **end**
6 Form an undirected graph $\hat{G}$ on $n$ nodes with edges $\{(i, j) : |\hat{A}_{ij}| \geq \eta/2, i < j\}$.

---

**Theorem 1.** *Let $\mathcal{D}(A, \theta)$ be an unknown $n$-variable Ising model distribution with dependency graph $G$. Suppose that the $\mathcal{D}(A, \theta)$ has width $\lambda(A, \theta) \leq \lambda$. Given $\rho \in (0, 1)$ and $\epsilon > 0$, if the number of i.i.d. samples satisfies $N = O(\lambda^2 \exp(12\lambda) \ln(n/\rho)/\epsilon^4)$, then with probability at least $1 - \rho$, Algorithm 1 produces $\hat{A}$ that satisfies*

$$\max_{i,j \in [n]} |A_{ij} - \hat{A}_{ij}| \leq \epsilon. \qquad (6)$$

**Corollary 1.** *In the setup of Theorem 1, suppose that the Ising model distribution $\mathcal{D}(A, \theta)$ has minimum edge weight $\eta(A, \theta) \geq \eta > 0$. If we set $\epsilon < \eta/2$ in (6), which corresponds to sample complexity $N = O(\lambda^2 \exp(12\lambda) \ln(n/\rho)/\eta^4)$, then with probability at least $1 - \rho$, Algorithm 1 recovers the dependency graph, i.e., $\hat{G} = G$.*

## 2.2 Learning pairwise graphical models over general alphabet

**Definition 2.** *Let $k$ be the alphabet size. Let $\mathcal{W} = \{W_{ij} \in \mathbb{R}^{k \times k} : i \neq j \in [n]\}$ be a set of weight matrices satisfying $W_{ij} = W_{ji}^T$. Without loss of generality, we assume that every row (and column) vector of $W_{ij}$ has zero mean. Let $\Theta = \{\theta_i \in \mathbb{R}^k : i \in [n]\}$ be a set of external field vectors. Then the $n$-variable pairwise graphical model $\mathcal{D}(\mathcal{W}, \Theta)$ is a distribution over $[k]^n$ where*

$$\mathbb{P}_{Z \sim \mathcal{D}(\mathcal{W},\Theta)}[Z = z] \propto \exp\left(\sum_{1 \leq i < j \leq n} W_{ij}(z_i, z_j) + \sum_{i \in [n]} \theta_i(z_i)\right). \qquad (7)$$

*The dependency graph of $\mathcal{D}(\mathcal{W}, \Theta)$ is an undirected graph $G = (V, E)$, with vertices $V = [n]$ and edges $E = \{(i, j) : W_{ij} \neq 0\}$. The width of $\mathcal{D}(\mathcal{W}, \Theta)$ is defined as*

$$\lambda(\mathcal{W}, \Theta) = \max_{i,a} \left(\sum_{j \neq i} \max_{b \in [k]} |W_{ij}(a, b)| + |\theta_i(a)|\right). \qquad (8)$$

*We define $\eta(\mathcal{W}, \Theta) = \min_{(i,j) \in E} \max_{a,b} |W_{ij}(a, b)|$.*

**Remark (centered rows and columns).** The assumption that $W_{ij}$ has centered rows and columns (i.e., $\sum_b W_{ij}(a, b) = 0$ and $\sum_a W_{ij}(a, b) = 0$ for any $a, b \in [k]$) is without loss of generality (see Fact 8.2 in (Klivans and Meka, 2017)). If the $a$-th row of $W_{ij}$ is not centered, i.e., $\sum_b W_{ij}(a, b) \neq 0$, we can define $W'_{ij}(a, b) = W_{ij}(a, b) - \sum_b W_{ij}(a, b)/k$ and $\theta'_i(a) = \theta_i(a) + \sum_b W_{ij}(a, b)/k$, and notice that $\mathcal{D}(\mathcal{W}, \Theta) = \mathcal{D}(\mathcal{W}', \Theta')$. Because the sets of matrices with centered rows and columns (i.e., $\{M \in \mathbb{R}^{k \times k} : \sum_b M(a, b) = 0, \forall a \in [k]\}$ and $\{M \in \mathbb{R}^{k \times k} : \sum_a M(a, b) = 0, \forall b \in [k]\}$) are two linear subspaces, alternatively projecting $W_{ij}$ onto the two sets will converge to the intersection of the two subspaces (Von Neumann, 1949). As a result, the condition of centered rows and columns is necessary for recovering the underlying weight matrices, since otherwise different parameters can give the same distribution. Note that in the case of $k = 2$, Definition 2 is the same as Definition 1 for Ising models. To see their connection, simply define $W_{ij} \in \mathbb{R}^{2 \times 2}$ as follows: $W_{ij}(1, 1) = W_{ij}(2, 2) = A_{ij}, W_{ij}(1, 2) = W_{ij}(2, 1) = -A_{ij}$.

For a pairwise graphical model distribution $\mathcal{D}(\mathcal{W}, \Theta)$, the conditional distribution of any variable (when restricted to a pair of values) given all the other variables follows a logistic function, as shown in Fact 2. This is analogous to Fact 1 for the Ising model distribution.

**Fact 2.** *Let $Z \sim \mathcal{D}(\mathcal{W}, \Theta)$ and $Z \in [k]^n$. For any $i \in [n]$, any $\alpha \neq \beta \in [k]$, and any $x \in [k]^{n-1}$,*

$$\mathbb{P}[Z_i = \alpha | Z_i \in \{\alpha, \beta\}, Z_{-i} = x] = \sigma(\sum_{j \neq i}(W_{ij}(\alpha, x_j) - W_{ij}(\beta, x_j)) + \theta_i(\alpha) - \theta_i(\beta)). \qquad (9)$$

Given $N$ i.i.d. samples $\{z^1, \cdots, z^N\}$, where $z^m \in [k]^n \sim \mathcal{D}(\mathcal{W}, \Theta)$ for $m \in [N]$, the goal is to estimate matrices $W_{ij}$ for all $i \neq j \in [n]$. For ease of notation and without loss of generality, let us consider the $n$-th variable. Now the goal is to estimate matrices $W_{nj}$ for all $j \in [n-1]$.

To use Fact 2, fix a pair of values $\alpha \neq \beta \in [k]$, let $S$ be the set of samples satisfying $z_n \in \{\alpha, \beta\}$. We next transform the samples in $S$ to $\{(x^t, y^t)\}_{t=1}^{|S|}$ as follows: $x^t = \mathrm{OneHotEncode}([z_{-n}^t, 1]) \in \{0, 1\}^{n \times k}, y^t = 1$ if $z_n^t = \alpha$, and $y^t = -1$ if $z_n^t = \beta$. Here $\mathrm{OneHotEncode}(\cdot) : [k]^n \to \{0, 1\}^{n \times k}$ is a function that maps a value $t \in [k]$ to the standard basis vector $e_t \in \{0, 1\}^k$, where $e_t$ has a single 1 at the $t$-th entry. For each sample $(x, y)$ in the set $S$, Fact 2 implies that $\mathbb{P}[y = 1|x] = \sigma(\langle w^*, x \rangle)$, where $w^* \in \mathbb{R}^{n \times k}$ satisfies

$$w^*(j, :) = W_{nj}(\alpha, :) - W_{nj}(\beta, :), \forall j \in [n-1]; \quad w^*(n, :) = [\theta_n(\alpha) - \theta_n(\beta), 0, ..., 0]. \qquad (10)$$

Suppose that the width of $\mathcal{D}(\mathcal{W}, \Theta)$ satisfies $\lambda(\mathcal{W}, \Theta) \leq \lambda$, then $w^*$ defined in (10) satisfies $\|w^*\|_{2,1} \leq 2\lambda\sqrt{k}$, where $\|w^*\|_{2,1} := \sum_j \|w^*(j, :)\|_2$. We can now form an $\ell_{2,1}$-constrained logistic regression over the samples in $S$:

$$w^{\alpha,\beta} \in \underset{w \in \mathbb{R}^{n \times k}}{\arg\min} \frac{1}{|S|} \sum_{t=1}^{|S|} \ln(1 + e^{-y^t \langle w, x^t \rangle}) \qquad \text{s.t. } \|w\|_{2,1} \leq 2\lambda\sqrt{k}. \qquad (11)$$

Let $w^{\alpha,\beta}$ be a minimizer of (11). We then create a new matrix $U^{\alpha,\beta} \in \mathbb{R}^{n \times k}$ by centering the first $n-1$ rows of $w^{\alpha,\beta}$:

$$U^{\alpha,\beta}(j,b) = w^{\alpha,\beta}(j,b) - \frac{1}{k}\sum_{a \in [k]} w^{\alpha,\beta}(j,a), \quad \forall j \in [n-1], \ \forall b \in [k]; \tag{12}$$

$$U^{\alpha,\beta}(n,b) = w^{\alpha,\beta}(n,b) + \frac{1}{k}\sum_{j \in [n-1], a \in [k]} w^{\alpha,\beta}(j,a), \quad \forall b \in [k].$$

Since each row of the $x$ matrix in (11) is a standard basis vector (i.e., all zeros except a single one), $\langle U^{\alpha,\beta}, x \rangle = \langle w^{\alpha,\beta}, x \rangle$, which implies that $U^{\alpha,\beta}$ is also a minimizer of (11).

The key step in our proof (see (30) in Appendix D) is to show that given enough samples, the obtained $U^{\alpha,\beta} \in \mathbb{R}^{n \times k}$ matrix is close to $w^*$ defined in (10). Specifically, we will prove that

$$|W_{nj}(\alpha,b) - W_{nj}(\beta,b) - U^{\alpha,\beta}(j,b)| \leq \epsilon, \quad \forall j \in [n-1], \ \forall \alpha,\beta,b \in [k]. \tag{13}$$

Recall that our goal is to estimate the original matrices $W_{nj}$ for all $j \in [n-1]$. Summing (13) over $\beta \in [k]$ and using the fact that $\sum_\beta W_{nj}(\beta,b) = 0$ gives

$$|W_{nj}(\alpha,b) - \frac{1}{k}\sum_{\beta \in [k]} U^{\alpha,\beta}(j,b)| \leq \epsilon, \quad \forall j \in [n-1], \ \forall \alpha,b \in [k]. \tag{14}$$

In other words, $\hat{W}_{nj}(\alpha,:) = \sum_{\beta \in [k]} U^{\alpha,\beta}(j,:)/k$ is a good estimate of $W_{nj}(\alpha,:)$.

The above algorithm is given in Algorithm 2. Suppose that $\eta(\mathcal{W},\Theta) \geq \eta$, once we obtain the estimates $\hat{W}_{ij}$, the last step is to form a graph by keeping the edge $(i,j)$ that satisfies $\max_{a,b}|\hat{W}_{ij}(a,b)| \geq \eta/2$.

---

**Algorithm 2:** Learning a pairwise graphical model via $\ell_{2,1}$-constrained logistic regression

---

**Input:** alphabet size $k$; $N$ i.i.d. samples $\{z^1, \cdots, z^N\}$, where $z^m \in [k]^n$ for $m \in [N]$; an upper bound on $\lambda(\mathcal{W},\Theta) \leq \lambda$; a lower bound on $\eta(\mathcal{W},\Theta) \geq \eta > 0$.
**Output:** $\hat{W}_{ij} \in \mathbb{R}^{k \times k}$ for all $i \neq j \in [n]$; an undirected graph $\hat{G}$ on $n$ nodes.

1 **for** $i \leftarrow 1$ **to** $n$ **do**
2      **for** *each pair* $\alpha \neq \beta \in [k]$ **do**
3          $S \leftarrow \{z^m, m \in [N] : z_i^m \in \{\alpha,\beta\}\}$
4          $\forall z^t \in S, x^t \leftarrow \text{OneHotEncode}([z_{-i}^t, 1]), y^t \leftarrow 1$ if $z_i^t = \alpha; y^t \leftarrow -1$ if $z_i^t = \beta$
5          $w^{\alpha,\beta} \leftarrow \arg\min_{w \in \mathbb{R}^{n \times k}} \frac{1}{|S|}\sum_{t=1}^{|S|} \ln(1 + e^{-y^t\langle w,x^t\rangle})$    s.t. $\|w\|_{2,1} \leq 2\lambda\sqrt{k}$
6          Define $U^{\alpha,\beta} \in \mathbb{R}^{n \times k}$ by centering the first $n-1$ rows of $w^{\alpha,\beta}$ (see (12)).
7      **end**
8      **for** $j \in [n]\backslash i$ *and* $\alpha \in [k]$ **do**
9          $\hat{W}_{ij}(\alpha,:) \leftarrow \frac{1}{k}\sum_{\beta \in [k]} U^{\alpha,\beta}(\tilde{j},:)$, where $\tilde{j} = j$ if $j < i$ and $\tilde{j} = j - 1$ if $j > i$.
10      **end**
11 **end**
12 Form graph $\hat{G}$ on $n$ nodes with edges $\{(i,j) : \max_{a,b}|\hat{W}_{ij}(a,b)| \geq \eta/2, i < j\}$.

---

**Theorem 2.** *Let $\mathcal{D}(\mathcal{W},\Theta)$ be an $n$-variable pairwise graphical model distribution with width $\lambda(\mathcal{W},\Theta) \leq \lambda$. Given $\rho \in (0,1)$ and $\epsilon > 0$, if the number of i.i.d. samples satisfies $N = O(\lambda^2 k^4 \exp(14\lambda)\ln(nk/\rho)/\epsilon^4)$, then with probability at least $1 - \rho$, Algorithm 2 produces $\hat{W}_{ij} \in \mathbb{R}^{k \times k}$ that satisfies*

$$|W_{ij}(a,b) - \hat{W}_{ij}(a,b)| \leq \epsilon, \quad \forall i \neq j \in [n], \ \forall a,b \in [k]. \tag{15}$$

**Corollary 2.** *In the setup of Theorem 2, suppose that the pairwise graphical model distribution $\mathcal{D}(\mathcal{W},\Theta)$ satisfies $\eta(\mathcal{W},\Theta) \geq \eta > 0$. If we set $\epsilon < \eta/2$ in (15), which corresponds to sample complexity $N = O(\lambda^2 k^4 \exp(14\lambda)\ln(nk/\rho)/\eta^4)$, then with probability at least $1 - \rho$, Algorithm 2 recovers the dependency graph, i.e., $\hat{G} = G$.*

**Remark ($\ell_{2,1}$ versus $\ell_1$ constraint).** The $w^* \in \mathbb{R}^{n \times k}$ matrix defined in (10) satisfies $\|w^*\|_{2,1} \le 2\lambda\sqrt{k}$ and $\|w^*\|_1 \le 2\lambda k$. Instead of solving the $\ell_{2,1}$-constrained logistic regression defined in (11), we could solve an $\ell_1$-constrained logistic regression with $\|w\|_1 \le 2\lambda k$. This additional $\sqrt{k}$ dependence in the constraint (i.e., $2\lambda k$ versus $2\lambda\sqrt{k}$) will lead to a worse sample complexity $\tilde{O}(k^5)$.

**Remark (dependence on the alphabet size).** A simple lower bound of the sample complexity is $\Omega(k^2)$. To see why, consider a graph with two nodes (i.e., $n = 2$). Let $W$ be a $k$-by-$k$ weight matrix between the two nodes, defined as follows: $W(1,1) = W(2,2) = 1$, $W(1,2) = W(2,1) = -1$, and $W(i,j) = 0$ otherwise. This definition satisfies the condition that every row and column is centered (Definition 2). Besides, we have $\lambda = 1$ and $\eta = 1$, which means that the two quantities do not scale in $k$. To distinguish $W$ from the zero matrix, we need to observe samples in the set $\{(1,1),(2,2),(1,2),(2,1)\}$. This requires $\Omega(k^2)$ samples because any specific sample $(a,b)$ (where $a \in [k]$ and $b \in [k]$) has a probability of approximately $1/k^2$ to show up.

## 2.3 Learning pairwise graphical models in $\tilde{O}(n^2)$ time

Our results so far assume that the $\ell_1$-constrained logistic regression (in Algorithm 1) and the $\ell_{2,1}$-constrained logistic regression (in Algorithm 2) is *exactly* solved. This would require $\tilde{O}(n^4)$ complexity if an interior-point based method is used (Koh et al., 2007). Our key result in this section is Theorem 3, which says that the statistical guarantees in Theorem 1 and 2 still hold if the constrained logistic regression is only *approximately* solved.

**Theorem 3** (Informal). *Suppose that the constrained logistic regression in Algorithm 1 and 2 is optimized by the mirror descent method given in Appendix J. Given $\rho \in (0,1)$ and $\epsilon > 0$, if the number of mirror descent iterations satisfies $T = O(\lambda^2 k^3 \exp(O(\lambda)) \ln(n)/\epsilon^4)$, then (6) and (15) still hold with probability at least $1 - \rho$. The time and space complexity of Algorithm 1 is $O(TNn^2)$ and $O(TN + n^2)$. The time and space complexity of Algorithm 2 is $O(TNn^2k^2)$ and $O(TN + n^2k^2)$.*

Proof of Theorem 3 requires bounding $\|\bar{w} - w^*\|_\infty$ (where $\bar{w}$ is the value after $T$ mirror descent iterations). This is non-trivial because we are in the high-dimensional regime as the number of samples $N = O(\ln(n))$, the empirical loss functions in (11) and (4) are not strongly convex. Due to the space limit, more details of this section can be found in Appendix I, J and K.

Note that $\tilde{O}(n^2)$ is an efficient time complexity for graph recovery over $n$ nodes. Previous structural learning algorithms of Ising models require either $\tilde{O}(n^2)$ complexity (Bresler, 2015; Klivans and Meka, 2017) or a worse time complexity (Ravikumar et al., 2010; Vuffray et al., 2016).

It is possible to improve the time complexity given in Theorem 3 (especially the dependence on $\epsilon$ and $\lambda$), by using stochastic or accelerated versions of mirror descent algorithms (instead of the batch version given in Appendix J). In fact, the Sparsitron algorithm proposed by Klivans and Meka (2017) can be seen as an online mirror descent algorithm for optimizing the $\ell_1$-constrained logistic regression (see Algorithm 3 in Appendix J). Furthermore, Algorithm 1 and 2 can be parallelized as every node has an independent regression problem.

We would like to remark that our goal here is not to give the fastest first-order optimization algorithm. Instead, our goal is to provably show that it is possible to run Algorithm 1 and Algorithm 2 in $\tilde{O}(n^2)$ time without affecting the original statistical guarantees.

## 3 Proof outline

We give a proof outline for Theorem 1. The proof of Theorem 2 follows a similar outline. Let $D$ be a distribution over $\{-1,1\}^n \times \{-1,1\}$, where $(x,y) \sim D$ satisfies $\mathbb{P}[y = 1|x] = \sigma(\langle w^*, x \rangle)$. Let $\mathcal{L}(w) = \mathbb{E}_{(x,y) \sim D} \ln(1 + e^{-y\langle w,x \rangle})$ and $\hat{\mathcal{L}}(w) = \sum_{i=1}^N \ln(1 + e^{-y^i \langle w, x^i \rangle})/N$ be the expected and empirical logistic loss. Suppose $\|w^*\|_1 \le 2\lambda$. Let $\hat{w} \in \arg\min_w \hat{\mathcal{L}}(w)$ s.t. $\|w\|_1 \le 2\lambda$. Our goal is to prove that $\|\hat{w} - w^*\|_\infty$ is small when the samples are constructed from an Ising model distribution.

Our proof can be summarized in three steps:

1. If the number of samples satisfies $N = O(\lambda^2 \ln(n/\rho)/\gamma^2)$, then $\mathcal{L}(\hat{w}) - \mathcal{L}(w^*) \leq O(\gamma)$. This is obtained using a sharp generalization bound when $\|w\|_1 \leq 2\lambda$ and $\|x\|_\infty \leq 1$ (see Lemma 8 in Appendix E).

2. For any $w$, we show that $\mathcal{L}(w) - \mathcal{L}(w^*) \geq \mathbb{E}_x[\sigma(\langle w, x \rangle) - \sigma(\langle w^*, x \rangle)]^2$ (see Lemma 10 and Lemma 9 in Appendix E). Hence, Step 1 implies that $\mathbb{E}_x[\sigma(\langle \hat{w}, x \rangle) - \sigma(\langle w^*, x \rangle)]^2 \leq O(\gamma)$ (see Lemma 1 in Appendix B).

3. We now use a result from (Klivans and Meka, 2017) (see Lemma 5 in Appendix B), which says that if the samples are from an Ising model and if $\gamma = O(\epsilon^2 \exp(-6\lambda))$, then $\mathbb{E}_x[\sigma(\langle \hat{w}, x \rangle) - \sigma(\langle w^*, x \rangle)]^2 \leq O(\gamma)$ implies that $\|\hat{w} - w^*\|_\infty \leq \epsilon$. The required number of samples is $N = O(\lambda^2 \ln(n/\rho)/\gamma^2) = O(\lambda^2 \exp(12\lambda) \ln(n/\rho)/\epsilon^4)$.

For the general setting with non-binary alphabet (i.e., Theorem 2), the proof is similar to that of Theorem 1. The main difference is that we need to use a sharp generalization bound when $\|w\|_{2,1} \leq 2\lambda\sqrt{k}$ and $\|x\|_{2,\infty} \leq 1$ (see Lemma 11 in Appendix E). This would give us Lemma 2 in Appendix B which bounds the squared distance between the two sigmoid functions. The last step is to use Lemma 6 to bound the infinity norm between the two weight matrices.

## 4 Experiments

In both of the simulations below, the external field is set to be zero. Sampling is done via exactly computing the distribution. We implement the algorithm in Matlab. All experiments are done using a personal desktop. Source code can be found at `https://github.com/wushanshan/GraphLearn`.

**Learning Ising models.** In Figure 1 we construct a diamond-shape graph and plot the incoherence value at Node 1. This value becomes bigger than 1 (and hence violates the incoherence condition in (Ravikumar et al., 2010)) when we increase the graph size $n$ and edge weight $a$. We then run 100 times of Algorithm 1 and plot the fraction of runs that exactly recovers the underlying graph structure. In each run we generate a different set of samples. The result shown in Figure 1 is consistent with our analysis and also indicates that our conditions for graph recovery are weaker than those in (Ravikumar et al., 2010).

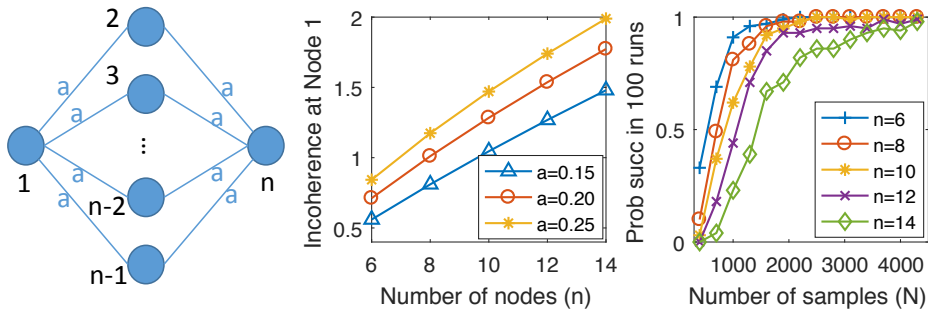

Figure 1: **Left**: The graph structure used in this simulation. It has $n$ nodes and $2(n-2)$ edges. Every edge has the same weight $a > 0$. **Middle**: Incoherence value at Node 1. The incoherence condition required by (Ravikumar et al., 2010) is violated for $n \geq 10$ and $a \geq 0.2$. **Right**: We simulate 100 runs of Algorithm 1 for edge weight $a = 0.2$ across different $n$ values.

**Learning general pairwise graphical models.** We compare our algorithm (Algorithm 2) with the Sparsitron algorithm in (Klivans and Meka, 2017) on a two-dimensional 3-by-3 grid (shown in Figure 2). We test two alphabet sizes in the experiments: $k = 4, 6$. For each value of $k$, we simulate both algorithms 100 runs, and in each run we generate random $W_{ij}$ matrices with entries $\pm 0.2$. To ensure that each row (as well as each column) of $W_{ij}$ is centered (Definition 2), we randomly choose $W_{ij}$ between two options: for example, if $k = 2$, then $W_{ij} = [0.2, -0.2; -0.2, 0.2]$ or $W_{ij} = [-0.2, 0.2; 0.2, -0.2]$. As shown in the Figure 2, our algorithm requires fewer samples for successfully recovering the graphs. More details about this experiment can be found in Appendix L.

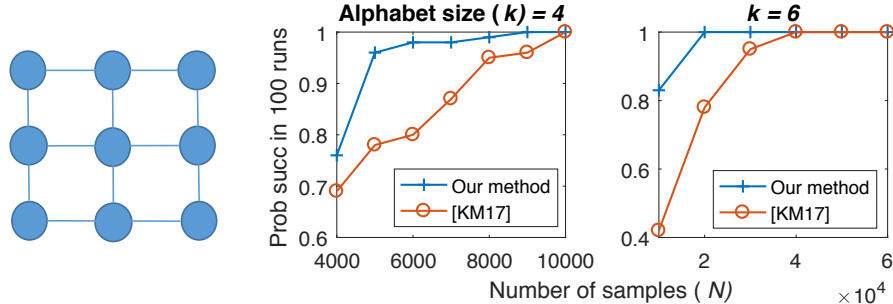

Figure 2: **Left**: A two-dimensional 3-by-3 grid graph used in the simulation. **Middle and right**: Our algorithm needs fewer samples than the Sparsitron algorithm (Klivans and Meka, 2017) for graph recovery.

## 5 Conclusion

The main contribution of this paper is to show that an existing and popular algorithm (i.e., group-sparse regularized logistic regression) actually gives the state-of-the-art performance (in a setting where alternative algorithms are being proposed). Specifically, we have shown that the $\ell_{2,1}$-constrained logistic regression can recover the Markov graph of any discrete pairwise graphical model from i.i.d. samples. For the special case of Ising model, the algorithm reduces to the $\ell_1$-constrained logistic regression. This algorithm has better sample complexity than the previous state-of-the-art result ($k^4$ versus $k^5$), and can be efficiently optimized in $\tilde{O}(n^2)$ time. One interesting direction for future work is to see if the $1/\eta^4$ dependency in the sample complexity can be improved. It is also interesting to see a thorough empirical evaluation of different structural learning algorithms.

Another interesting direction is to consider MRFs with higher-order interactions. Intuitively, it should not be difficult to prove that $\ell_1$-constrained logistic regression can recover the structure of binary $t$-wise MRFs. One can prove it by combining results from Section 7 of (Klivans and Meka, 2017) and the following fact: the Sparsitron algorithm can be viewed as an online mirror descent algorithm that approximately solves an $\ell_1$-constrained logistic regression. This observation is actually the starting point of our paper. For higher-order MRFs with non-binary alphabet, we conjecture that similar result can be proved for group-sparse regularized logistic regression. Extending the current proof/method to higher-order MRFs is definitely an interesting direction for future research.

## 6 Acknowledgements

This research has been supported by NSF Grants 1302435, 1564000, and 1618689, DMS 1723052, CCF 1763702, AF 1901292 and research gifts by Google, Western Digital and NVIDIA.

## Footnotes

[2]Theorem 8.4 in (Klivans and Meka, 2017) has a typo. The correct dependence should be $k^5$ instead of $k^3$. In Section 8 of (Klivans and Meka, 2017), after re-writing the conditional distribution as a sigmoid function, the weight vector $w$ is a vector of length $(n-1)k + 1$. Their derivation uses an incorrect bound $\|w\|_1 \leq 2\lambda$, while it should be $\|w\|_1 \leq 2k\lambda$. This gives rise to an additional $k^2$ factor on the final sample complexity.

[3]It may be possible to prove a similar result for the *regularized* version of the optimization problem using techniques from (Negahban et al., 2012). One needs to prove that the objective function satisfies restricted strong convexity (RSC) when the samples are from a graphical model distribution (Vuffray et al., 2016; Lokhov et al., 2018). It would be interesting to see if the proof presented in this paper is related to the RSC condition.

[4]This improvement essentially comes from the fact that we are using an $\ell_{2,1}$ norm constraint instead of an $\ell_1$ norm constraint for learning general (i.e., non-binary) pairwise graphical models (see our remark after Theorem 2). The Sparsitron algorithm proposed by Klivans and Meka (2017) learns a $\ell_1$-constrained generalized linear model. This $\ell_1$-constraint gives rise to a $k^5$ dependency for learning non-binary pairwise graphical models.

[5]In an independent and concurrent work, Vuffray et al. (2019) generalize the Interaction Screening algorithm (Vuffray et al., 2016). Their sample complexity is $O(k^4 \hat{\gamma}^4 \exp(12\lambda) \ln(nk)/\eta^4)$ for learning pairwise non-binary graphical models (see Corollary 4 in their paper), where $\hat{\gamma}$ is an upper bound on the $\ell_1$ norm of the node-wise weight vectors. Since $\hat{\gamma}$ can scale as $k^2 \lambda$, their dependence on $k$ can be much worse than ours.

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
