[Supplementary Material · NIPS19_graphicalModel_camera_ready_app.pdf]

# A   Related work on learning Ising models

For the special case of learning Ising models (i.e., binary variables), we compare the sample complexity among different graph recovery algorithms in Table 2. Note that most of the results here (Ravikumar et al., 2010; Bresler, 2015; Vuffray et al., 2016; Lokhov et al., 2018; Rigollet and Hütter, 2017) are only for learning Ising models. Therefore, we do not list them in Table 1.

| Paper | Assumptions | Sample complexity ($N$) |
|---|---|---|
| Information-theoretic lower bound (Santhanam and Wainright, 2012) | 1. Model width $\leq \lambda$, and $\lambda \geq 1$ <br> 2. Degree $\leq d$ <br> 3. Minimum edge weight $\geq \eta > 0$ <br> 4. External field $= 0$ | $\max\{\frac{\ln(n)}{2\eta \tanh(\eta)},$ <br> $\frac{d}{8}\ln(\frac{n}{8d}),$ <br> $\frac{\exp(\lambda)\ln(nd/4-1)}{4\eta d \exp(\eta)}\}$ |
| $\ell_1$-regularized logistic regression (Ravikumar et al., 2010) | $Q^*$ is the Fisher information matrix, <br> $S$ is set of neighbors of a given variable. <br> 1. Dependency: $\exists\, C_{\min} > 0$ such that <br> $\quad$ eigenvalues of $Q^*_{SS} \geq C_{\min}$ <br> 2. Incoherence: $\exists\, \alpha \in (0,1]$ such that <br> $\quad \|Q^*_{S^c S}(Q^*_{SS})^{-1}\|_\infty \leq 1-\alpha$ <br> 3. Regularization parameter: <br> $\quad \lambda_N \geq \frac{16(2-\alpha)}{\alpha}\sqrt{\frac{\ln(n)}{N}}$ <br> 4. Minimum edge weight $\geq 10\sqrt{d}\lambda_N/C_{\min}$ <br> 5. External field $= 0$ <br> 6. Probability of success $\geq 1 - 2e^{-O(\lambda_N^2 N)}$ | $O(d^3 \ln(n))$ |
| Greedy algorithm (Bresler, 2015) | 1. Model width $\leq \lambda$ <br> 2. Degree $\leq d$ <br> 3. Minimum edge weight $\geq \eta > 0$ <br> 4. Probability of success $\geq 1 - \rho$ | $O(\exp(\frac{\exp(O(d\lambda))}{\eta^{O(1)}})\ln(\frac{n}{\rho}))$ |
| Interaction Screening (Vuffray et al., 2016) | 1. Model width $\leq \lambda$ <br> 2. Degree $\leq d$ <br> 3. Minimum edge weight $\geq \eta > 0$ <br> 4. Regularization parameter $= 4\sqrt{\frac{\ln(3n^2/\rho)}{N}}$ <br> 5. Probability of success $\geq 1 - \rho$ | $O(\max\{d, \frac{1}{\eta^2}\}$ <br> $d^3\exp(6\lambda)\ln(\frac{n}{\rho}))$ |
| $\ell_1$-regularized logistic regression (Lokhov et al., 2018) | 1. Model width $\leq \lambda$ <br> 2. Degree $\leq d$ <br> 3. Minimum edge weight $\geq \eta > 0$ <br> 4. Regularization parameter $O(\sqrt{\frac{\ln(n^2/\rho)}{N}})$ <br> 5. Probability of success $\geq 1 - \rho$ | $O(\max\{d, \frac{1}{\eta^2}\}$ <br> $d^3\exp(8\lambda)\ln(\frac{n}{\rho}))$ |
| $\ell_1$-constrained logistic regression (Rigollet and Hütter, 2017) | 1. Model width $\leq \lambda$ <br> 2. Minimum edge weight $\geq \eta > 0$ <br> 3. Probability of success $\geq 1 - \rho$ | $O(\frac{\lambda^2 \exp(8\lambda)}{\eta^4}\ln(\frac{n}{\rho}))$ |
| Sparsitron (Klivans and Meka, 2017) | 1. Model width $\leq \lambda$ <br> 2. Minimum edge weight $\geq \eta > 0$ <br> 3. Probability of success $\geq 1 - \rho$ | $O(\frac{\lambda^2 \exp(12\lambda)}{\eta^4}\ln(\frac{n}{\rho\eta}))$ |
| $\ell_1$-constrained logistic regression (**this paper**) | 1. Model width $\leq \lambda$ <br> 2. Minimum edge weight $\geq \eta > 0$ <br> 3. Probability of success $\geq 1 - \rho$ | $O(\frac{\lambda^2 \exp(12\lambda)}{\eta^4}\ln(\frac{n}{\rho}))$ |

Table 2: Comparison of the sample complexity required for graph recovery of an Ising model. The second column lists the assumptions in their analysis. Given $\lambda$ and $\eta$, the degree is bounded by $d \leq \lambda/\eta$, with equality achieved when every edge has the same weight and there is no external field.

As mentioned in the Introduction, Ravikumar et al. (2010) consider $\ell_1$-regularized logistic regression for learning Ising models in the high-dimensional setting. They require incoherence assumptions that

ensure, via conditions on sub-matrices of the Fisher information matrix, that sparse predictors of each node are hard to confuse with a false set. Their analysis obtains significantly better sample complexity compared to what is possible when these extra assumptions are not imposed (Bento and Montanari, 2009). Others have also considered $\ell_1$-regularization (Lee et al., 2007; Yuan and Lin, 2007; Banerjee et al., 2008; Jalali et al., 2011; Yang et al., 2012; Aurell and Ekeberg, 2012) for structure learning of Markov random fields but they all require certain assumptions about the graphical model and hence their methods do not work for general graphical models. The analysis of (Ravikumar et al., 2010) is of essentially the same convex program as this work (except that we have an additional thresholding procedure). The main difference is that they obtain a better sample guarantee but require significantly more restrictive assumptions.

In the general setting with no restrictions on the model, Santhanam and Wainwright (2012) provide an information-theoretic lower bound on the number of samples needed for graph recovery. This lower bound depends logarithmically on $n$, and exponentially on the width $\lambda$, and (somewhat inversely) on the minimum edge weight $\eta$. We will find these general broad trends, but with important differences, in the other algorithms as well.

Bresler (2015) provides a greedy algorithm and shows that it can learn with sample complexity that grows logarithmically in $n$, but *doubly* exponentially in the width $\lambda$ and also exponentially in $1/\eta$. It is thus suboptimal with respect to its dependence on $\lambda$ and $\eta$.

Vuffray et al. (2016) propose a new convex program (i.e. different from logistic regression), and for this they are able to show a single-exponential dependence on $\lambda$. There is also low-order polynomial dependence on $\lambda$ and $1/\eta$. Note that given $\lambda$ and $\eta$, the degree is bounded by $d \leq \lambda/\eta$ (the equality is achieved when every edge has the same weight and there is no external field). Therefore, their sample complexity can scale as worse as $1/\eta^5$. Later, the same authors (Lokhov et al., 2018) prove a similar result for the $\ell_1$-regularized logistic regression using essentially the same proof technique as (Vuffray et al., 2016).

Rigollet and Hütter (2017) analyze the $\ell_1$-constrained logistic regression for learning Ising models. Their sample complexity[6] has a better dependence on $1/\eta$ ($1/\eta^4$ vs $1/\eta^5$) than (Lokhov et al., 2018). It would be interesting to see if their analysis can be extended to the $\ell_{2,1}$-constrained logistic regression. Naïve extension may give a sample complexity exponential in the alphabet size[7].

In this paper, we analyze the $\ell_{2,1}$-constrained logistic regression for learning discrete pairwise graphical models with general alphabet. Our proof uses generalization bound, and is different from (Lokhov et al., 2018; Rigollet and Hütter, 2017). For learning Ising models (shown in Table 2), our sample complexity matches that of (Klivans and Meka, 2017). For learning non-binary pairwise graphical models (shown in Table 1), our sample complexity improves the state-of-the-art result.

## B    Supporting lemmas

In this section, we list the lemmas that will be used in proving our main theorems: Theorem 1 and 2. Proofs of the two theorems are given in Appendix C and D. Proofs of the supporting lemmas are given in Appendix E to H.

Lemma 1 and Lemma 2 are the key results in our proof. They essentially say that given enough samples, solving the corresponding constrained logistic regression problem will provide a prediction $\sigma(\langle \hat{w}, x \rangle)$ close to the true $\sigma(\langle w^*, x \rangle)$ in terms of their expected squared distance.

**Lemma 1.** *Let $\mathcal{D}$ be a distribution on $\{-1, 1\}^n \times \{-1, 1\}$ where for $(X, Y) \sim \mathcal{D}$, $\mathbb{P}[Y = 1 | X = x] = \sigma(\langle w^*, x \rangle)$. We assume that $\|w^*\|_1 \leq 2\lambda$ for a known $\lambda \geq 0$. Given $N$ i.i.d. samples $\{(x^i, y^i)\}_{i=1}^N$, let $\hat{w}$ be any minimizer of the following $\ell_1$-constrained logistic regression problem:*

$$\hat{w} \in \operatorname*{arg\,min}_{w \in \mathbb{R}^n} \frac{1}{N} \sum_{i=1}^N \ln(1 + e^{-y^i \langle w, x^i \rangle}) \quad \text{s.t.} \ \|w\|_1 \leq 2\lambda. \tag{16}$$

*Given $\rho \in (0, 1)$ and $\epsilon > 0$, if the number of samples satisfies $N = O(\lambda^2 \ln(n/\rho)/\epsilon^2)$, then with probability at least $1 - \rho$ over the samples, $\mathbb{E}_{(x,y)\sim\mathcal{D}}[(\sigma(\langle w^*, x\rangle) - \sigma(\langle \hat{w}, x\rangle))^2] \leq \epsilon$.*

**Lemma 2.** *Let $\mathcal{D}$ be a distribution on $\mathcal{X} \times \{-1, 1\}$, where $\mathcal{X} = \{x \in \{0, 1\}^{n\times k} : \|x\|_{2,\infty} \leq 1\}$. Furthermore, $(X, Y) \sim \mathcal{D}$ satisfies $\mathbb{P}[Y = 1|X = x] = \sigma(\langle w^*, x\rangle)$, where $w^* \in \mathbb{R}^{n\times k}$. We assume that $\|w^*\|_{2,1} \leq 2\lambda\sqrt{k}$ for a known $\lambda \geq 0$. Given $N$ i.i.d. samples $\{(x^i, y^i)\}_{i=1}^N$ from $\mathcal{D}$, let $\hat{w}$ be any minimizer of the following $\ell_{2,1}$-constrained logistic regression problem:*

$$\hat{w} \in \operatorname*{arg\,min}_{w\in\mathbb{R}^{n\times k}} \frac{1}{N}\sum_{i=1}^{N} \ln(1 + e^{-y^i\langle w, x^i\rangle}) \quad \text{s.t. } \|w\|_{2,1} \leq 2\lambda\sqrt{k}. \tag{17}$$

*Given $\rho \in (0, 1)$ and $\epsilon > 0$, if the number of samples satisfies $N = O(\lambda^2 k(\ln(n/\rho))/\epsilon^2)$, then with probability at least $1 - \rho$ over the samples, $\mathbb{E}_{(x,y)\sim\mathcal{D}}[(\sigma(\langle w^*, x\rangle) - \sigma(\langle \hat{w}, x\rangle))^2] \leq \epsilon$.*

The proofs of Lemma 1 and Lemma 2 are given in Appendix E. Note that in the setup of both lemmas, we form a pair of dual norms for $x$ and $w$, e.g., $\|x\|_{2,\infty}$ and $\|w\|_{2,1}$ in Lemma 2, and $\|x\|_{\infty}$ and $\|w\|_1$ in Lemma 1. This duality allows us to use a sharp generalization bound with a sample complexity that scales logarithmic in the dimension (see Lemma 8 and Lemma 11 in Appendix E).

Definition 3 defines a $\delta$-unbiased distribution. This notion of $\delta$-unbiasedness is proposed by Klivans and Meka (2017).

**Definition 3.** *Let $S$ be the alphabet set, e.g., $S = \{-1, 1\}$ for Ising model and $S = [k]$ for an alphabet of size $k$. A distribution $\mathcal{D}$ on $S^n$ is $\delta$-unbiased if for $X \sim \mathcal{D}$, any $i \in [n]$, and any assignment $x \in S^{n-1}$ to $X_{-i}$, $\min_{\alpha\in S}(\mathbb{P}[X_i = \alpha|X_{-i} = x]) \geq \delta$.*

For a $\delta$-unbiased distribution, any of its marginal distribution is also $\delta$-unbiased (see Lemma 3).

**Lemma 3.** *Let $\mathcal{D}$ be a $\delta$-unbiased distribution on $S^n$, where $S$ is the alphabet set. For $X \sim \mathcal{D}$, any $i \in [n]$, the distribution of $X_{-i}$ is also $\delta$-unbiased.*

Lemma 4 describes the $\delta$-unbiased property of graphical models. This property has been used in the previous papers (e.g., (Klivans and Meka, 2017; Bresler, 2015)).

**Lemma 4.** *Let $\mathcal{D}(\mathcal{W}, \Theta)$ be a pairwise graphical model distribution with alphabet size $k$ and width $\lambda(\mathcal{W}, \Theta)$. Then $\mathcal{D}(\mathcal{W}, \Theta)$ is $\delta$-unbiased with $\delta = e^{-2\lambda(\mathcal{W},\Theta)}/k$. Specifically, an Ising model distribution $\mathcal{D}(A, \theta)$ is $e^{-2\lambda(A,\theta)}/2$-unbiased.*

In Lemma 1 and Lemma 2, we give a sample complexity bound for achieving a small $\ell_2$ error between $\sigma(\langle \hat{w}, x\rangle)$ and $\sigma(\langle w^*, x\rangle)$. The following two lemmas show that if the sample distribution is $\delta$-unbiased, then a small $\ell_2$ error implies a small distance between $\hat{w}$ and $w^*$.

**Lemma 5.** *Let $\mathcal{D}$ be a $\delta$-unbiased distribution on $\{-1, 1\}^n$. Suppose that for two vectors $u, w \in \mathbb{R}^n$ and $\theta', \theta'' \in \mathbb{R}$, $\mathbb{E}_{X\sim\mathcal{D}}[(\sigma(\langle w, X\rangle + \theta') - \sigma(\langle u, X\rangle + \theta''))^2] \leq \epsilon$, where $\epsilon < \delta e^{-2\|w\|_1 - 2|\theta'| - 6}$. Then $\|w - u\|_{\infty} \leq O(1) \cdot e^{\|w\|_1 + |\theta'|} \cdot \sqrt{\epsilon/\delta}$.*

**Lemma 6.** *Let $\mathcal{D}$ be a $\delta$-unbiased distribution on $[k]^n$. For $X \sim \mathcal{D}$, let $\tilde{X} \in \{0, 1\}^{n\times k}$ be the one-hot encoded $X$. Let $u, w \in \mathbb{R}^{n\times k}$ be two matrices satisfying $\sum_a u(i, a) = 0$ and $\sum_a w(i, a) = 0$, for $i \in [n]$. Suppose that for some $u, w$ and $\theta', \theta'' \in \mathbb{R}$, we have $\mathbb{E}_{X\sim\mathcal{D}}[(\sigma(\langle w, \tilde{X}\rangle + \theta') - \sigma(\langle u, \tilde{X}\rangle + \theta''))^2] \leq \epsilon$, where $\epsilon < \delta e^{-2\|w\|_{\infty,1} - 2|\theta'| - 6}$. Then[8] $\|w - u\|_{\infty} \leq O(1) \cdot e^{\|w\|_{\infty,1} + |\theta'|} \cdot \sqrt{\epsilon/\delta}$.*

The proofs of Lemma 5 and Lemma 6 can be found in (Klivans and Meka, 2017) (see Claim 8.6 and Lemma 4.3 in their paper). We give a slightly different proof of these two lemmas in Appendix H.

## C  Proof of Theorem 1

We first restate Theorem 1 and then give the proof.

**Theorem.** *Let $\mathcal{D}(A, \theta)$ be an unknown $n$-variable Ising model distribution with dependency graph $G$. Suppose that the $\mathcal{D}(A, \theta)$ has width $\lambda(A, \theta) \leq \lambda$. Given $\rho \in (0, 1)$ and $\epsilon > 0$, if the number of i.i.d. samples satisfies $N = O(\lambda^2 \exp(12\lambda) \ln(n/\rho)/\epsilon^4)$, then with probability at least $1 - \rho$, Algorithm 1 produces $\hat{A}$ that satisfies*

$$\max_{i,j \in [n]} |A_{ij} - \hat{A}_{ij}| \leq \epsilon. \tag{18}$$

*Proof.* For ease of notation, we consider the $n$-th variable. The goal is to prove that Algorithm 1 is able to recover the $n$-th row of the true weight matrix $A$. Specifically, we will show that if the number samples satisfies $N = O(\lambda^2 \exp(O(\lambda)) \ln(n/\rho)/\epsilon^4)$, then with probability as least $1 - \rho/n$,

$$\max_{j \in [n]} |A_{nj} - \hat{A}_{nj}| \leq \epsilon. \tag{19}$$

We then use a union bound to conclude that with probability as least $1 - \rho$, $\max_{i,j \in [n]} |A_{ij} - \hat{A}_{ij}| \leq \epsilon$.

Let $Z \sim \mathcal{D}(A, \theta)$, $X = [Z_{-n}, 1] = [Z_1, Z_2, \cdots, Z_{n-1}, 1] \in \{-1, 1\}^n$, and $Y = Z_n \in \{-1, 1\}$. By Fact 1, $\mathbb{P}[Y = 1 | X = x] = \sigma(\langle w^*, x \rangle)$, where $w^* = 2[A_{n1}, \cdots, A_{n(n-1)}, \theta_n]$. Further, $\|w^*\|_1 \leq 2\lambda$. Let $\hat{w}$ be the solution of the $\ell_1$-constrained logistic regression problem defined in (4).

By Lemma 1, if the number of samples satisfies $N = O(\lambda^2 \ln(n/\rho)/\gamma^2)$, then with probability at least $1 - \rho/n$, we have

$$\mathbb{E}_X[(\sigma(\langle w^*, X \rangle) - \sigma(\langle \hat{w}, X \rangle))^2] \leq \gamma. \tag{20}$$

By Lemma 4, $Z_{-n} \in \{-1, 1\}^{n-1}$ is $\delta$-unbiased (Definition 3) with $\delta = e^{-2\lambda}/2$. By Lemma 5, if $\gamma < C_1 \delta e^{-4\lambda}$ for some constant $C_1 > 0$, then (20) implies that

$$\|w^*_{1:(n-1)} - \hat{w}_{1:(n-1)}\|_\infty \leq O(1) \cdot e^{2\lambda} \cdot \sqrt{\gamma/\delta}. \tag{21}$$

Note that $w^*_{1:(n-1)} = 2[A_{n1}, \cdots, A_{n(n-1)}]$ and $\hat{w}_{1:(n-1)} = 2[\hat{A}_{n1}, \cdots, \hat{A}_{n(n-1)}]$. Let $\gamma = C_2 \delta e^{-4\lambda} \epsilon^2$ for some constant $C_2 > 0$ and $\epsilon \in (0, 1)$, (21) then implies that

$$\max_{j \in [n]} |A_{nj} - \hat{A}_{nj}| \leq \epsilon. \tag{22}$$

The number of samples needed is $N = O(\lambda^2 \ln(n/\rho)/\gamma^2) = O(\lambda^2 e^{12\lambda} \ln(n/\rho)/\epsilon^4)$.

We have proved that (19) holds with probability at least $1 - \rho/n$. Using a union bound over all $n$ variables gives that with probability as least $1 - \rho$, $\max_{i,j \in [n]} |A_{ij} - \hat{A}_{ij}| \leq \epsilon$. $\square$

# D  Proof of Theorem 2

The following lemma will be used in the proof.

**Lemma 7.** *Let $Z \sim \mathcal{D}$, where $\mathcal{D}$ is a $\delta$-unbiased distribution on $[k]^n$. Given $\alpha \neq \beta \in [k]$, conditioned on $Z_n \in \{\alpha, \beta\}$, $Z_{-n} \in [k]^{n-1}$ is also $\delta$-unbiased.*

*Proof.* For any $i \in [n-1]$, $a \in [k]$, and $x \in [k]^{n-2}$, we have

$$\mathbb{P}[Z_i = a | Z_{[n] \setminus \{i,n\}} = x, Z_n \in \{\alpha, \beta\}]$$

$$= \frac{\mathbb{P}[Z_i = a, Z_{[n] \setminus \{i,n\}} = x, Z_n = \alpha] + \mathbb{P}[Z_i = a, Z_{[n] \setminus \{i,n\}} = x, Z_n = \beta]}{\mathbb{P}[Z_{[n] \setminus \{i,n\}} = x, Z_n = \alpha] + \mathbb{P}[Z_{[n] \setminus \{i,n\}} = x, Z_n = \beta]}$$

$$\overset{(a)}{\geq} \min\left(\frac{\mathbb{P}[Z_i = a, Z_{[n] \setminus \{i,n\}} = x, Z_n = \alpha]}{\mathbb{P}[Z_{[n] \setminus \{i,n\}} = x, Z_n = \alpha]}, \frac{\mathbb{P}[Z_i = a, Z_{[n] \setminus \{i,n\}} = x, Z_n = \beta]}{\mathbb{P}[Z_{[n] \setminus \{i,n\}} = x, Z_n = \beta]}\right)$$

$$= \min(\mathbb{P}[Z_i = a | Z_{[n] \setminus \{i,n\}} = x, Z_n = \alpha], \mathbb{P}[Z_i = a | Z_{[n] \setminus \{i,n\}} = x, Z_n = \beta])$$

$$\overset{(b)}{\geq} \delta. \tag{23}$$

where (a) follows from the fact that $(a + b)/(c + d) \geq \min(a/c, b/d)$ for $a, b, c, d > 0$, (b) follows from the fact that $Z$ is $\delta$-unbiased. $\square$

Now we are ready to prove Theorem 2, which is restated below.

**Theorem.** *Let $\mathcal{D}(\mathcal{W}, \Theta)$ be an $n$-variable pairwise graphical model distribution with width $\lambda(\mathcal{W}, \Theta) \leq \lambda$ and alphabet size $k$. Given $\rho \in (0, 1)$ and $\epsilon > 0$, if the number of i.i.d. samples satisfies $N = O(\lambda^2 k^4 \exp(14\lambda) \ln(nk/\rho)/\epsilon^4)$, then with probability at least $1 - \rho$, Algorithm 2 produces $\hat{W}_{ij} \in \mathbb{R}^{k \times k}$ that satisfies*

$$|W_{ij}(a, b) - \hat{W}_{ij}(a, b)| \leq \epsilon, \quad \forall i \neq j \in [n], \ \forall a, b \in [k]. \tag{24}$$

*Proof.* To ease notation, let us consider the $n$-th variable (i.e., set $i = n$ inside the first "for" loop of Algorithm 2). The proof directly applies to other variables. We will prove the following result: if the number of samples $N = O(\lambda^2 k^4 \exp(14\lambda) \ln(nk/\rho)/\epsilon^4)$, then with probability at least $1 - \rho/n$, the $U^{\alpha,\beta} \in \mathbb{R}^{n \times k}$ matrices produced by Algorithm 2 satisfies

$$|W_{nj}(\alpha, b) - W_{nj}(\beta, b) - U^{\alpha,\beta}(j, b)| \leq \epsilon, \quad \forall j \in [n-1], \ \forall \alpha, \beta, b \in [k]. \tag{25}$$

Suppose that (25) holds, summing over $\beta \in [k]$ and using the fact that $\sum_\beta W_{nj}(\beta, b) = 0$ gives

$$\left|W_{nj}(\alpha, b) - \frac{1}{k} \sum_{\beta \in [k]} U^{\alpha,\beta}(j, b)\right| \leq \epsilon, \quad \forall j \in [n-1], \ \forall \alpha, b \in [k]. \tag{26}$$

Theorem 2 then follows by taking a union bound over the $n$ variables.

The only thing left is to prove (25). Now fix a pair of $\alpha, \beta \in [k]$, let $N^{\alpha,\beta}$ be the number of samples such that the $n$-th variable is either $\alpha$ or $\beta$. By Lemma 2 and Fact 2, if $N^{\alpha,\beta} = O(\lambda^2 k \ln(n/\rho')/\gamma^2)$, then with probability at least $1 - \rho'$, the minimizer of the $\ell_{2,1}$ constrained logistic regression $w^{\alpha,\beta} \in \mathbb{R}^{n \times k}$ satisfies

$$\mathbb{E}_X[(\sigma(\langle w^*, X \rangle) - \sigma(\langle w^{\alpha,\beta}, X \rangle))^2] \leq \gamma. \tag{27}$$

Recall that $X \in \{0, 1\}^{n \times k}$ is the one-hot encoding of the vector $[Z_{-n}, 1] \in [k]^n$, where $Z \sim \mathcal{D}(\mathcal{W}, \Theta)$ and $Z_n \in \{\alpha, \beta\}$. Besides, $w^* \in \mathbb{R}^{n \times k}$ satisfies

$$w^*(j, :) = W_{nj}(\alpha, :) - W_{nj}(\beta, :), \ \forall j \in [n-1]; \quad w^*(n, :) = [\theta_n(\alpha) - \theta_n(\beta), 0, \cdots, 0]. \tag{28}$$

Let $U^{\alpha,\beta} \in \mathbb{R}^{n \times k}$ be formed by centering the first $n - 1$ rows of $w^{\alpha,\beta}$. Since each row of $X$ is a standard basis vector (i.e., all 0's except a single 1), $\langle U^{\alpha,\beta}, X \rangle = \langle w^{\alpha,\beta}, X \rangle$. Hence, (27) implies

$$\mathbb{E}_X[(\sigma(\langle w^*, X \rangle) - \sigma(\langle U^{\alpha,\beta}, X \rangle))^2] \leq \gamma. \tag{29}$$

By Lemma 4, we know that $Z \sim \mathcal{D}(\mathcal{W}, \Theta)$ is $\delta$-unbiased with $\delta = e^{-2\lambda}/k$. By Lemma 7, conditioned on $Z_n \in \{\alpha, \beta\}$, $Z_{-n}$ is also $\delta$-unbiased. Hence, the condition of Lemma 6 holds. Applying Lemma 6 to (29), we get that if $N^{\alpha,\beta} = O(\lambda^2 k^3 \exp(12\lambda) \ln(n/\rho'))/\epsilon^4)$, the following holds with probability at least $1 - \rho'$:

$$|W_{nj}(\alpha, b) - W_{nj}(\beta, b) - U^{\alpha,\beta}(j, b)| \leq \epsilon, \ \forall j \in [n-1], \ \forall b \in [k]. \tag{30}$$

So far we have proved that (25) holds for a fixed $(\alpha, \beta)$ pair. This requires that $N^{\alpha,\beta} = O(\lambda^2 k^3 \exp(12\lambda) \ln(n/\rho'))/\epsilon^4)$. Recall that $N^{\alpha,\beta}$ is the number of samples that the $n$-th variable takes $\alpha$ or $\beta$. We next derive the number of total samples needed in order to have $N^{\alpha,\beta}$ samples for a given $(\alpha, \beta)$ pair. Since $\mathcal{D}(\mathcal{W}, \Theta)$ is $\delta$-unbiased with $\delta = e^{-2\lambda(\mathcal{W},\Theta)}/k$, for $Z \sim \mathcal{D}(\mathcal{W}, \Theta)$, we have $\mathbb{P}[Z_n \in \{\alpha, \beta\}|Z_{-n}] \geq 2\delta$, and hence $\mathbb{P}[Z_n \in \{\alpha, \beta\}] \geq 2\delta$. By the Chernoff bound, if the total number of samples satisfies $N = O(N^{\alpha,\beta}/\delta + \log(1/\rho'')/\delta)$, then with probability at least $1 - \rho''$, we have $N^{\alpha,\beta}$ samples for a given $(\alpha, \beta)$ pair.

To ensure that (30) holds for all $(\alpha, \beta)$ pairs with probability at least $1 - \rho/n$, we can set $\rho' = \rho/(nk^2)$ and $\rho'' = \rho/(nk^2)$ and take a union bound over all $(\alpha, \beta)$ pairs. The total number of samples required is $N = O(\lambda^2 k^4 \exp(14\lambda) \ln(nk/\rho)/\epsilon^4)$.

We have shown that (25) holds for the $n$-th variable with probability at least $1 - \rho/n$. By the discussion at the beginning of the proof, Theorem 2 then follows by a union bound over the $n$ variables. $\square$

# E   Proof of Lemma 1 and Lemma 2

The proof of Lemma 1 relies on the following lemmas. The first lemma is a generalization error bound for any Lipschitz loss of linear functions with bounded $\|w\|_1$ and $\|x\|_\infty$.

**Lemma 8.** *(see, e.g., Corollary 4 of (Kakade et al., 2009) and Theorem 26.15 of (Shalev-Shwartz and Ben-David, 2014)) Let $\mathcal{D}$ be a distribution on $\mathcal{X} \times \mathcal{Y}$, where $\mathcal{X} = \{x \in \mathbb{R}^n : \|x\|_\infty \leq X_\infty\}$, and $\mathcal{Y} = \{-1, 1\}$. Let $\ell : \mathbb{R} \to \mathbb{R}$ be a loss function with Lipschitz constant $L_\ell$. Define the expected loss $\mathcal{L}(w)$ and the empirical loss $\hat{\mathcal{L}}(w)$ as*

$$\mathcal{L}(w) = \mathop{\mathbb{E}}_{(x,y)\sim\mathcal{D}} \ell(y \langle w, x \rangle), \quad \hat{\mathcal{L}}(w) = \frac{1}{N}\sum_{i=1}^{N} \ell(y^i \langle w, x^i \rangle), \tag{31}$$

*where $\{x^i, y^i\}_{i=1}^{N}$ are i.i.d. samples from distribution $\mathcal{D}$. Define $\mathcal{W} = \{w \in \mathbb{R}^n : \|w\|_1 \leq W_1\}$. Then with probability at least $1 - \rho$ over the samples, we have that for all $w \in \mathcal{W}$,*

$$\mathcal{L}(w) \leq \hat{\mathcal{L}}(w) + 2L_\ell X_\infty W_1 \sqrt{\frac{2\ln(2n)}{N}} + L_\ell X_\infty W_1 \sqrt{\frac{2\ln(2/\rho)}{N}}. \tag{32}$$

**Lemma 9.** *(Pinsker's inequality) Let $D_{KL}(a\|b) := a\ln(a/b) + (1-a)\ln((1-a)/(1-b))$ denote the KL-divergence between two Bernoulli distributions $(a, 1-a)$, $(b, 1-b)$ with $a, b \in [0, 1]$. Then*

$$(a - b)^2 \leq \frac{1}{2}D_{KL}(a\|b). \tag{33}$$

**Lemma 10.** *Let $\mathcal{D}$ be a distribution on $\mathcal{X} \times \{-1, 1\}$. For $(X, Y) \sim \mathcal{D}$, $\mathbb{P}[Y = 1|X = x] = \sigma(\langle w^*, x \rangle)$, where $\sigma(x) = 1/(1 + e^{-x})$ is the sigmoid function. Let $\mathcal{L}(w)$ be the expected logistic loss:*

$$\mathcal{L}(w) = \mathop{\mathbb{E}}_{(x,y)\sim\mathcal{D}} \ln(1 + e^{-y\langle w, x\rangle}) = \mathop{\mathbb{E}}_{(x,y)\sim\mathcal{D}}[-\frac{y+1}{2}\ln(\sigma(\langle w, x\rangle)) - \frac{1-y}{2}\ln(1 - \sigma(\langle w, x\rangle))]. \tag{34}$$

*Then for any $w$, we have*

$$\mathcal{L}(w) - \mathcal{L}(w^*) = \mathop{\mathbb{E}}_{(x,y)\sim\mathcal{D}}[D_{KL}(\sigma(\langle w^*, x\rangle)\|\sigma(\langle w, x\rangle))], \tag{35}$$

*where $D_{KL}(a\|b) := a\ln(a/b) + (1-a)\ln((1-a)/(1-b))$ denotes the KL-divergence between two Bernoulli distributions $(a, 1-a)$, $(b, 1-b)$ with $a, b \in [0, 1]$.*

*Proof.* Simply plugging in the definition of the expected logistic loss $\mathcal{L}(\cdot)$ gives

$$\mathcal{L}(w) - \mathcal{L}(w^*) = \mathop{\mathbb{E}}_{(x,y)\sim\mathcal{D}}[-\frac{y+1}{2}\ln(\sigma(\langle w, x\rangle)) - \frac{1-y}{2}\ln(1 - \sigma(\langle w, x\rangle))]$$

$$+ \mathop{\mathbb{E}}_{(x,y)\sim\mathcal{D}}[\frac{y+1}{2}\ln(\sigma(\langle w^*, x\rangle)) + \frac{1-y}{2}\ln(1 - \sigma(\langle w^*, x\rangle))]$$

$$= \mathop{\mathbb{E}}_{x}\mathop{\mathbb{E}}_{y|x}[-\frac{y+1}{2}\ln(\sigma(\langle w, x\rangle)) - \frac{1-y}{2}\ln(1 - \sigma(\langle w, x\rangle))]$$

$$+ \mathop{\mathbb{E}}_{x}\mathop{\mathbb{E}}_{y|x}[\frac{y+1}{2}\ln(\sigma(\langle w^*, x\rangle)) + \frac{1-y}{2}\ln(1 - \sigma(\langle w^*, x\rangle))]$$

$$\overset{(a)}{=} \mathop{\mathbb{E}}_{x}[-\sigma(\langle w^*, x\rangle)\ln(\sigma(\langle w, x\rangle)) - (1 - \sigma(\langle w^*, x\rangle))\ln(1 - \sigma(\langle w, x\rangle))]$$

$$+ \mathop{\mathbb{E}}_{x}[\sigma(\langle w^*, x\rangle)\ln(\sigma(\langle w^*, x\rangle)) + (1 - \sigma(\langle w^*, x\rangle))\ln(1 - \sigma(\langle w^*, x\rangle))]$$

$$= \mathop{\mathbb{E}}_{x}\left[\sigma(\langle w^*, x\rangle)\ln\left(\frac{\sigma(\langle w^*, x\rangle)}{\sigma(\langle w, x\rangle)}\right) + (1 - \sigma(\langle w^*, x\rangle))\ln\left(\frac{1 - \sigma(\langle w^*, x\rangle)}{1 - \sigma(\langle w, x\rangle)}\right)\right]$$

$$= \mathop{\mathbb{E}}_{(x,y)\sim\mathcal{D}}[D_{KL}(\sigma(\langle w^*, x\rangle)\|\sigma(\langle w, x\rangle))],$$

where (a) follows from the fact that

$$E_{y|x}[y] = 1 \cdot \mathbb{P}[y = 1|x] + (-1) \cdot \mathbb{P}[y = -1|x] = 2\sigma(\langle w^*, x\rangle) - 1.$$

$\square$

We are now ready to prove Lemma 1 (which is restated below):

**Lemma.** *Let $\mathcal{D}$ be a distribution on $\{-1, 1\}^n \times \{-1, 1\}$ where for $(X, Y) \sim \mathcal{D}$, $\mathbb{P}[Y = 1 | X = x] = \sigma(\langle w^*, x \rangle)$. We assume that $\|w^*\|_1 \leq 2\lambda$ for a known $\lambda \geq 0$. Given $N$ i.i.d. samples $\{(x^i, y^i)\}_{i=1}^N$, let $\hat{w}$ be any minimizer of the following $\ell_1$-constrained logistic regression problem:*

$$\hat{w} \in \arg \min_{w \in \mathbb{R}^n} \frac{1}{N} \sum_{i=1}^N \ln(1 + e^{-y^i \langle w, x^i \rangle}) \quad \text{s.t. } \|w\|_1 \leq 2\lambda. \tag{36}$$

*Given $\rho \in (0, 1)$ and $\epsilon > 0$, suppose that $N = O(\lambda^2(\ln(n/\rho))/\epsilon^2)$, then with probability at least $1 - \rho$ over the samples, we have that $\mathbb{E}_{(x,y)\sim\mathcal{D}}[(\sigma(\langle w^*, x \rangle) - \sigma(\langle \hat{w}, x \rangle))^2] \leq \epsilon$.*

*Proof.* We first apply Lemma 8 to the setup of Lemma 1. The loss function $\ell(z) = \ln(1 + e^{-z})$ defined above has Lipschitz constant $L_\ell = 1$. The input sample $x \in \{-1, 1\}^n$ satisfies $\|x\|_\infty \leq 1$. Let $\mathcal{W} = \{w \in \mathbb{R}^{n \times k} : \|w\|_1 \leq 2\lambda\}$. According to Lemma 8, with probability at least $1 - \rho/2$ over the draw of the training set, we have that for all $w \in \mathcal{W}$,

$$\mathcal{L}(w) \leq \hat{\mathcal{L}}(w) + 4\lambda \sqrt{\frac{2\ln(2n)}{N}} + 2\lambda \sqrt{\frac{2\ln(4/\rho)}{N}}. \tag{37}$$

where $\mathcal{L}(w) = \mathbb{E}_{(x,y)\sim\mathcal{D}} \ln(1 + e^{-y\langle w, x \rangle})$ and $\hat{\mathcal{L}}(w) = \sum_{i=1}^N \ln(1 + e^{-y^i \langle w, x^i \rangle})/N$ are the expected loss and empirical loss.

Let $N = C \cdot \lambda^2 \ln(8n/\rho)/\epsilon^2$ for a global constant $C$, then (37) implies that with probability at least $1 - \rho/2$,

$$\mathcal{L}(w) \leq \hat{\mathcal{L}}(w) + \epsilon, \text{ for all } w \in \mathcal{W}. \tag{38}$$

We next prove a concentration result for $\hat{\mathcal{L}}(w^*)$. Here $w^*$ is the true regression vector and is assumed to be fixed. First notice that $\ln(1 + e^{-y\langle w^*, x \rangle})$ is bounded because $|y \langle w^*, x \rangle| \leq 2\lambda$. Besides, the $\ln(1 + e^{-z})$ has Lipschitz 1, so $|\ln(1 + e^{-2\lambda}) - \ln(1 + e^{2\lambda})| \leq 4\lambda$. Hoeffding's inequality gives that $\mathbb{P}[\hat{\mathcal{L}}(w^*) - \mathcal{L}(w^*) \geq t] \leq e^{-2Nt^2/(4\lambda)^2}$. Let $N = C' \cdot \lambda^2 \ln(2/\rho)/\epsilon^2$ for a global constant $C'$, then with probability at least $1 - \rho/2$ over the samples,

$$\hat{\mathcal{L}}(w^*) \leq \mathcal{L}(w^*) + \epsilon. \tag{39}$$

Then the following holds with probability at least $1 - \rho$:

$$\mathcal{L}(\hat{w}) \overset{(a)}{\leq} \hat{\mathcal{L}}(\hat{w}) + \epsilon \overset{(b)}{\leq} \hat{\mathcal{L}}(w^*) + \epsilon \overset{(c)}{\leq} \mathcal{L}(w^*) + 2\epsilon, \tag{40}$$

where (a) follows from (38), (b) follows from the fact $\hat{w}$ is the minimizer of $\hat{\mathcal{L}}(w)$, and (c) follows from (39).

So far we have shown that $\mathcal{L}(\hat{w}) - \mathcal{L}(w^*) \leq 2\epsilon$ with probability at least $1 - \rho$. The last step is to lower bound $\mathcal{L}(\hat{w}) - \mathcal{L}(w^*)$ by $\mathbb{E}_{(x,y)\sim\mathcal{D}}(\sigma(\langle w^*, x \rangle) - \sigma(\langle w, x \rangle))^2$ using Lemma 9 and Lemma 10.

$$\mathbb{E}_{(x,y)\sim\mathcal{D}}(\sigma(\langle w^*, x \rangle) - \sigma(\langle w, x \rangle))^2 \overset{(d)}{\leq} \mathbb{E}_{(x,y)\sim\mathcal{D}} D_{KL}(\sigma(\langle w^*, x \rangle) \| \sigma(\langle w, x \rangle))/2$$

$$\overset{(e)}{=} (\mathcal{L}(\hat{w}) - \mathcal{L}(w^*))/2$$

$$\overset{(f)}{\leq} \epsilon,$$

where (d) follows from Lemma 9, (e) follows from Lemma 10, and (f) follows from (40). Therefore, we have that $\mathbb{E}_{(x,y)\sim\mathcal{D}}(\sigma(\langle w^*, x \rangle) - \sigma(\langle w, x \rangle))^2 \leq \epsilon$ with probability at least $1 - \rho$, if the number of samples satisfies $N = O(\lambda^2 \ln(n/\rho)/\epsilon^2)$. $\qquad\square$

The proof of Lemma 2 is identical to the proof of Lemma 1, except that it relies on the following generalization error bound for Lipschitz loss functions with bounded $\ell_{2,1}$-norm.

**Lemma 11.** *Let $\mathcal{D}$ be a distribution on $\mathcal{X} \times \mathcal{Y}$, where $\mathcal{X} = \{x \in \mathbb{R}^{n \times k} : \|x\|_{2,\infty} \leq X_{2,\infty}\}$, and $\mathcal{Y} = \{-1, 1\}$. Let $\ell : \mathbb{R} \to \mathbb{R}$ be a loss function with Lipschitz constant $L_\ell$. Define the expected loss $\mathcal{L}(w)$ and the empirical loss $\hat{\mathcal{L}}(w)$ as*

$$\mathcal{L}(w) = \mathbb{E}_{(x,y) \sim \mathcal{D}} \ell(y \langle w, x \rangle), \quad \hat{\mathcal{L}}(w) = \frac{1}{N} \sum_{i=1}^{N} \ell(y^i \langle w, x^i \rangle), \tag{41}$$

*where $\{x^i, y^i\}_{i=1}^{N}$ are i.i.d. samples from distribution $\mathcal{D}$. Define $\mathcal{W} = \{w \in \mathbb{R}^{n \times k} : \|w\|_{2,1} \leq W_{2,1}\}$. Then with probability at least $1 - \rho$ over the draw of $N$ samples, we have that for all $w \in \mathcal{W}$,*

$$\mathcal{L}(w) \leq \hat{\mathcal{L}}(w) + 2L_\ell X_{2,\infty} W_{2,1} \sqrt{\frac{6 \ln(n)}{N}} + L_\ell X_{2,\infty} W_{2,1} \sqrt{\frac{2 \ln(2/\rho)}{N}}. \tag{42}$$

Lemma 11 can be readily derived from the existing results. First, notice that the dual norm of $\|\cdot\|_{2,1}$ is $\|\cdot\|_{2,\infty}$. Using Corollary 14 in (Kakade et al., 2012), Theorem 1 in (Kakade et al., 2009), and the fact that $\|w\|_{2,q} \leq \|w\|_{2,1}$ for $q \geq 1$, we conclude that the Rademacher complexity of the function class $\mathcal{F} := \{x \to \langle w, x \rangle : \|w\|_{2,1} \leq W_{2,1}\}$ is at most $X_{2,\infty} W_{2,1} \sqrt{6 \ln(n)/N}$. We can then obtain the standard Rademacher-based generalization bound (see, e.g., (Bartlett and Mendelson, 2002) and Theorem 26.5 in (Shalev-Shwartz and Ben-David, 2014)) for bounded Lipschitz loss functions.

We omit the proof of Lemma 2 since it is the same as that of Lemma 1.

## F    Proof of Lemma 3

Lemma 3 is restated below.

**Lemma.** *Let $\mathcal{D}$ be a $\delta$-unbiased distribution on $S^n$, where $S$ is the alphabet set. For $X \sim \mathcal{D}$, any $i \in [n]$, the distribution of $X_{-i}$ is also $\delta$-unbiased.*

*Proof.* For any $j \neq i \in [n]$, any $a \in S$, and any $x \in S^{n-2}$, we have

$$\mathbb{P}[X_j = a | X_{[n] \setminus \{i,j\}} = x] = \sum_{b \in S} \mathbb{P}[X_j = a, X_i = b | X_{[n] \setminus \{i,j\}} = x]$$

$$= \sum_{b \in S} \mathbb{P}[X_i = b | X_{[n] \setminus \{i,j\}} = x] \cdot \mathbb{P}[X_j = a | X_i = b, X_{[n] \setminus \{i,j\}} = x]$$

$$\overset{(a)}{\geq} \delta \sum_{b \in S} \mathbb{P}[X_i = b | X_{[n] \setminus \{i,j\}} = x]$$

$$= \delta, \tag{43}$$

where (a) follows from the fact that $X \sim \mathcal{D}$ and $\mathcal{D}$ is a $\delta$-unbiased distribution. Since (43) holds for any $j \neq i \in [n]$, any $a \in S$, and any $x \in S^{n-2}$, by definition, the distribution of $X_{-i}$ is $\delta$-unbiased. $\qquad\square$

## G    Proof of Lemma 4

The lemma is restated below, followed by its proof.

**Lemma.** *Let $\mathcal{D}(\mathcal{W}, \Theta)$ be a pairwise graphical model distribution with alphabet size $k$ and width $\lambda(\mathcal{W}, \Theta)$. Then $\mathcal{D}(\mathcal{W}, \Theta)$ is $\delta$-unbiased with $\delta = e^{-2\lambda(\mathcal{W}, \Theta)}/k$. Specifically, an Ising model distribution $\mathcal{D}(A, \theta)$ is $e^{-2\lambda(A,\theta)}/2$-unbiased.*

*Proof.* Let $X \sim \mathcal{D}(\mathcal{W}, \Theta)$, and assume that $X \in [k]^n$. For any $i \in [n]$, any $a \in [k]$, and any $x \in [k]^{n-1}$, we have

$$\mathbb{P}[X_i = a | X_{-i} = x] = \frac{\exp(\sum_{j \neq i} W_{ij}(a, x_j) + \theta_i(a))}{\sum_{b \in [k]} \exp(\sum_{j \neq i} W_{ij}(b, x_j) + \theta_i(b))}$$

$$= \frac{1}{\sum_{b \in [k]} \exp(\sum_{j \neq i} (W_{ij}(b, x_j) - W_{ij}(a, x_j)) + \theta_i(b) - \theta_i(a))}$$

$$\overset{(a)}{\geq} \frac{1}{k \cdot \exp(2\lambda(\mathcal{W}, \Theta))} = e^{-2\lambda(\mathcal{W}, \Theta)}/k, \tag{44}$$

where (a) follows from the definition of model width. The lemma then follows (Ising model corresponds to the special case of $k = 2$). $\qquad\square$

## H  Proof of Lemma 5 and Lemma 6

The proof relies on the following basic property of the sigmoid function (see Claim 4.2 of (Klivans and Meka, 2017)):

$$|\sigma(a) - \sigma(b)| \geq e^{-|a|-3} \cdot \min(1, |a - b|), \quad \forall a, b \in \mathbb{R}. \tag{45}$$

We first prove Lemma 5 (which is restated below).

**Lemma.** *Let $\mathcal{D}$ be a $\delta$-unbiased distribution on $\{-1, 1\}^n$. Suppose that for two vectors $u, w \in \mathbb{R}^n$ and $\theta', \theta'' \in \mathbb{R}$, $\mathbb{E}_{X \sim \mathcal{D}}[(\sigma(\langle w, X \rangle + \theta') - \sigma(\langle u, X \rangle + \theta''))^2] \leq \epsilon$, where $\epsilon < \delta e^{-2\|w\|_1 - 2|\theta'| - 6}$. Then $\|w - u\|_\infty \leq O(1) \cdot e^{\|w\|_1 + |\theta'|} \cdot \sqrt{\epsilon/\delta}$.*

*Proof.* For any $i \in [n]$, any $X \in \{-1, 1\}^n$, let $X_i \in \{-1, 1\}$ be the $i$-th variable and $X_{-i} \in \{-1, 1\}^{n-1}$ be the $[n] \backslash \{i\}$ variables. Let $X^{i,+} \in \{-1, 1\}^n$ (respectively $X^{i,-}$) be the vector obtained from $X$ by setting $X_i = 1$ (respectively $X_i = -1$). Then we have

$$\epsilon \geq \mathbb{E}_{X \sim \mathcal{D}}[(\sigma(\langle w, X \rangle + \theta') - \sigma(\langle u, X \rangle + \theta''))^2]$$

$$= \mathbb{E}_{X_{-i}} \left[ \mathbb{E}_{X_i | X_{-i}} (\sigma(\langle w, X \rangle + \theta') - \sigma(\langle u, X \rangle + \theta''))^2 \right]$$

$$= \mathbb{E}_{X_{-i}} [(\sigma(\langle w, X^{i,+} \rangle + \theta') - \sigma(\langle u, X^{i,+} \rangle + \theta''))^2 \cdot \mathbb{P}[X_i = 1 | X_{-i}]$$

$$\qquad + (\sigma(\langle w, X^{i,-} \rangle + \theta') - \sigma(\langle u, X^{i,-} \rangle + \theta''))^2 \cdot \mathbb{P}[X_i = -1 | X_{-i}]]$$

$$\overset{(a)}{\geq} \delta \cdot \mathbb{E}_{X_{-i}} [(\sigma(\langle w, X^{i,+} \rangle + \theta') - \sigma(\langle u, X^{i,+} \rangle + \theta''))^2$$

$$\qquad + (\sigma(\langle w, X^{i,-} \rangle + \theta') - \sigma(\langle u, X^{i,-} \rangle + \theta''))^2]$$

$$\overset{(b)}{\geq} \delta e^{-2\|w\|_1 - 2|\theta'| - 6} \cdot \mathbb{E}_{X_{-i}} [\min(1, ((\langle w, X^{i,+} \rangle + \theta') - (\langle u, X^{i,+} \rangle + \theta''))^2)$$

$$\qquad + \min(1, ((\langle w, X^{i,-} \rangle + \theta') - (\langle u, X^{i,-} \rangle + \theta''))^2)]$$

$$\overset{(c)}{\geq} \delta e^{-2\|w\|_1 - 2|\theta'| - 6} \cdot \mathbb{E}_{X_{-i}} \min(1, (2w_i - 2u_i)^2/2)$$

$$\overset{(d)}{=} \delta e^{-2\|w\|_1 - 2|\theta'| - 6} \cdot \min(1, 2(w_i - u_i)^2). \tag{46}$$

Here (a) follows from the fact that $\mathcal{D}$ is a $\delta$-unbiased distribution, which implies that $\mathbb{P}[X_i = 1 | X_{-i}] \geq \delta$ and $\mathbb{P}[X_i = -1 | X_{-i}] \geq \delta$. Inequality (b) is obtained by substituting (45). Inequality (c) uses the following fact

$$\min(1, a^2) + \min(1, b^2) \geq \min(1, (a - b)^2/2), \forall a, b \in \mathbb{R}. \tag{47}$$

To see why (47) holds, note that if both $|a|, |b| \leq 1$, then (47) is true since $a^2 + b^2 \geq (a - b)^2/2$. Otherwise, (47) is true because the left-hand side is at least 1 while the right-hand side is at most 1. The last equality (d) follows from that $X_{-i}$ is independent of $\min(1, 2(w_i - u_i)^2)$.

Since $\epsilon < \delta e^{-2\|w\|_1 - 2|\theta'| - 6}$, (46) implies that $|w_i - u_i| \leq O(1) \cdot e^{\|w\|_1 + |\theta'|} \cdot \sqrt{\epsilon/\delta}$. Because (46) holds for any $i \in [n]$, we have that $\|w - u\|_\infty \leq O(1) \cdot e^{\|w\|_1 + |\theta'|} \cdot \sqrt{\epsilon/\delta}$. $\qquad\square$

We now prove Lemma 6 (which is restated below).

**Lemma.** *Let $\mathcal{D}$ be a $\delta$-unbiased distribution on $[k]^n$. For $X \sim \mathcal{D}$, let $\tilde{X} \in \{0,1\}^{n \times k}$ be the one-hot encoded $X$. Let $u, w \in \mathbb{R}^{n \times k}$ be two matrices satisfying $\sum_j u(i,j) = 0$ and $\sum_j w(i,j) = 0$ for $i \in [n]$. Suppose that for some $u, w$ and $\theta', \theta'' \in \mathbb{R}$, we have $\mathbb{E}_{X \sim \mathcal{D}}[(\sigma(\langle w, \tilde{X} \rangle + \theta') - \sigma(\langle u, \tilde{X} \rangle + \theta''))^2] \leq \epsilon$, where $\epsilon < \delta e^{-2\|w\|_{\infty,1} - 2|\theta'| - 6}$. Then $\|w - u\|_\infty \leq O(1) \cdot e^{\|w\|_{\infty,1} + |\theta'|} \cdot \sqrt{\epsilon/\delta}$.*

*Proof.* Fix an $i \in [n]$ and $a \neq b \in [k]$. Let $X^{i,a} \in [k]^n$ (respectively $X^{i,b}$) be the vector obtained from $X$ by setting $X_i = a$ (respectively $X_i = b$). Let $\tilde{X}^{i,a} \in \{0,1\}^{n \times k}$ be the one-hot encoding of $X^{i,a} \in [k]^n$. Then we have

$$
\begin{aligned}
\epsilon &\geq \mathop{\mathbb{E}}_{X \sim \mathcal{D}}[(\sigma(\langle w, \tilde{X} \rangle + \theta') - \sigma(\langle u, \tilde{X} \rangle + \theta''))^2] \\
&= \mathop{\mathbb{E}}_{X_{-i}}\left[\mathop{\mathbb{E}}_{X_i | X_{-i}} (\sigma(\langle w, \tilde{X} \rangle + \theta') - \sigma(\langle u, \tilde{X} \rangle + \theta''))^2\right] \\
&\geq \mathop{\mathbb{E}}_{X_{-i}}[(\sigma(\langle w, \tilde{X}^{i,a} \rangle + \theta') - \sigma(\langle u, \tilde{X}^{i,a} \rangle + \theta''))^2 \cdot \mathbb{P}[X_i = a | X_{-i}] \\
&\qquad + (\sigma(\langle w, \tilde{X}^{i,b} \rangle + \theta') - \sigma(\langle u, \tilde{X}^{i,b} \rangle + \theta''))^2 \cdot \mathbb{P}[X_i = b | X_{-i}]] \\
&\overset{(a)}{\geq} \delta e^{-2\|w\|_{\infty,1} - 2|\theta'| - 6} \cdot \mathop{\mathbb{E}}_{X_{-i}}[\min(1, ((\langle w, \tilde{X}^{i,a} \rangle + \theta') - (\langle u, \tilde{X}^{i,a} \rangle + \theta''))^2) \\
&\qquad + \min(1, ((\langle w, \tilde{X}^{i,b} \rangle + \theta') - (\langle u, \tilde{X}^{i,b} \rangle + \theta''))^2)] \\
&\overset{(b)}{\geq} \delta e^{-2\|w\|_{\infty,1} - 2|\theta'| - 6} \cdot \mathop{\mathbb{E}}_{X_{-i}} \min(1, ((w(i,a) - w(i,b)) - (u(i,a) - u(i,b)))^2/2) \\
&= \delta e^{-2\|w\|_{\infty,1} - 2|\theta'| - 6} \min(1, ((w(i,a) - w(i,b)) - (u(i,a) - u(i,b)))^2/2) \qquad (48)
\end{aligned}
$$

Here (a) follows from that $\mathcal{D}$ is a $\delta$-unbiased distribution and (45). Inequality (b) follows from (47). Because $\epsilon < \delta e^{-2\|w\|_{\infty,1} - 2|\theta'| - 6}$, (48) implies that

$$(w(i,a) - w(i,b)) - (u(i,a) - u(i,b)) \leq O(1) \cdot e^{\|w\|_{\infty,1} + |\theta'|} \cdot \sqrt{\epsilon/\delta}. \qquad (49)$$

$$(u(i,a) - u(i,b)) - (w(i,a) - w(i,b)) \leq O(1) \cdot e^{\|w\|_{\infty,1} + |\theta'|} \cdot \sqrt{\epsilon/\delta}. \qquad (50)$$

Since (49) and (50) hold for any $a \neq b \in [k]$, we can sum over $b \in [k]$ and use the fact that $\sum_j u(i,j) = 0$ and $\sum_j w(i,j) = 0$ to get

$$w(i,a) - u(i,a) = \frac{1}{k}\sum_b (w(i,a) - w(i,b)) - (u(i,a) - u(i,b)) \leq O(1) \cdot e^{\|w\|_{\infty,1} + |\theta'|} \cdot \sqrt{\epsilon/\delta}.$$

$$u(i,a) - w(i,a) = \frac{1}{k}\sum_b (u(i,a) - u(i,b)) - (w(i,a) - w(i,b)) \leq O(1) \cdot e^{\|w\|_{\infty,1} + |\theta'|} \cdot \sqrt{\epsilon/\delta}.$$

Therefore, we have $|w(i,a) - u(i,a)| \leq O(1) \cdot e^{\|w\|_{\infty,1} + |\theta'|} \cdot \sqrt{\epsilon/\delta}$, for any $i \in [n]$ and $a \in [k]$. $\qquad\square$

# I   Learning pairwise graphical models in $\tilde{O}(n^2)$ time

Our results so far assume that the $\ell_1$-constrained logistic regression (in Algorithm 1) and the $\ell_{2,1}$-constrained logistic regression (in Algorithm 2) can be solved exactly. This would require $\tilde{O}(n^4)$ complexity if an interior-point based method is used (Koh et al., 2007). The goal of this section is to reduce the runtime to $\tilde{O}(n^2)$ via first-order optimization method. Note that $\tilde{O}(n^2)$ is an efficient time complexity for graph recovery over $n$ nodes. Previous structural learning algorithms of Ising models require either $\tilde{O}(n^2)$ complexity ((Bresler, 2015; Klivans and Meka, 2017)) or a worse complexity

((Ravikumar et al., 2010; Vuffray et al., 2016) require $\tilde{O}(n^4)$ runtime). We would like to remark that our goal here is not to give the fastest first-order optimization algorithm (see our remark after Theorem 5). Instead, our contribution is to provably show that it is possible to run Algorithm 1 and Algorithm 2 in $\tilde{O}(n^2)$ time without affecting the original statistical guarantees.

To better exploit the problem structure[9], we use the mirror descent algorithm[10] with a properly chosen distance generating function (aka the mirror map). Following the standard mirror descent setup, we use negative entropy as the mirror map for $\ell_1$-constrained logistic regression and a scaled group norm for $\ell_{2,1}$-constrained logistic regression (see Section 5.3.3.2 and Section 5.3.3.3 in (Ben-Tal and Nemirovski, 2013) for more details). The pseudocode is given in Appendix J. The main advantage of mirror descent algorithm is that its convergence rate scales *logarithmically* in the dimension (see Lemma 12 in Appendix K). Specifically, let $\bar{w}$ be the output after $O(\ln(n)/\gamma^2)$ mirror descent iterations, then $\bar{w}$ satisfies

$$\hat{\mathcal{L}}(\bar{w}) - \hat{\mathcal{L}}(\hat{w}) \leq \gamma, \tag{51}$$

where $\hat{\mathcal{L}}(w) = \sum_{i=1}^{N} \ln(1 + e^{-y^i\langle w,x^i\rangle})/N$ is the empirical logistic loss, and $\hat{w}$ is the actual minimizer of $\hat{\mathcal{L}}(w)$. Since each mirror descent update requires $O(nN)$ time, where $N$ is the number of samples and scales as $O(\ln(n))$, and we have to solve $n$ regression problems (one for each variable in $[n]$), the total runtime scales as $\tilde{O}(n^2)$, which is our desired runtime.

There is still one problem left, that is, we have to show that $\|\bar{w} - w^*\|_\infty \leq \epsilon$ (where $w^*$ is the minimizer of the true loss $\mathcal{L}(w) = \mathbb{E}_{(x,y)\sim\mathcal{D}} \ln(1 + e^{-y\langle w,x\rangle})$) in order to conclude that Theorem 1 and 2 still hold when using mirror descent algorithms. Since $\hat{\mathcal{L}}(w)$ is not strongly convex, (51) alone does not necessarily imply that $\|\bar{w} - \hat{w}\|_\infty$ is small. Our key insight is that in the proof of Theorem 1 and 2, the definition of $\hat{w}$ (as a minimizer of $\hat{\mathcal{L}}(w)$) is only used to show that $\hat{\mathcal{L}}(\hat{w}) \leq \hat{\mathcal{L}}(w^*)$ (see inequality (b) of (40) in Appendix E). It is then possible to replace this step with (51) in the original proof, and prove that Theorem 1 and 2 still hold as long as $\gamma$ is small enough (see (60) in Appendix K).

Our key results in this section are Theorem 4 and Theorem 5, which show that Algorithm 1 and Algorithm 2 can run in $\tilde{O}(n^2)$ time without affecting the original statistical guarantees.

**Theorem 4.** *In the setup of Theorem 1, suppose that the $\ell_1$-constrained logistic regression in Algorithm 1 is optimized by the mirror descent method (Algorithm 3) given in Appendix J. Given $\rho \in (0,1)$ and $\epsilon > 0$, if the number of mirror descent iterations satisfies $T = O(\lambda^2 \exp(12\lambda) \ln(n)/\epsilon^4)$, and the number of i.i.d. samples satisfies $N = O(\lambda^2 \exp(12\lambda) \ln(n/\rho)/\epsilon^4)$, then (6) still holds with probability at least $1 - \rho$. The total time complexity of Algorithm 1 is $O(TNn^2)$.*

**Theorem 5.** *In the setup of Theorem 2, suppose that the $\ell_{2,1}$-constrained logistic regression in Algorithm 2 is optimized by the mirror descent method (Algorithm 4) given in Appendix J. Given $\rho \in (0,1)$ and $\epsilon > 0$, if the number of mirror descent iterations satisfies $T = O(\lambda^2 k^3 \exp(12\lambda) \ln(n)/\epsilon^4)$, and the number of i.i.d. samples satisfies $N = O(\lambda^2 k^4 \exp(14\lambda) \ln(nk/\rho)/\epsilon^4)$, then (15) still holds with probability at least $1 - \rho$. The total time complexity of Algorithm 2 is $O(TNn^2k^2)$.*

**Remark.** It is possible to improve the time complexity given in Theorem 4 and 5 (especially the dependence on $\epsilon$ and $\lambda$), by using stochastic or accelerated versions of mirror descent algorithms (instead of the batch version given in Appendix J). In fact, the Sparsitron algorithm proposed by Klivans and Meka (2017) can be seen as an online mirror descent algorithm for optimizing the $\ell_1$-constrained logistic regression (see Algorithm 3 in Appendix J). Furthermore, Algorithm 1 and 2 can be parallelized as the regression problem is defined separately for each variable.

## J Mirror descent algorithms for constrained logistic regression

Algorithm 3 gives a mirror descent algorithm for the following $\ell_1$-constrained logistic regression:

$$\min_{w \in \mathbb{R}^n} \frac{1}{N} \sum_{i=1}^{N} \ln(1 + e^{-y^i \langle w, x^i \rangle}) \qquad \text{s.t. } \|w\|_1 \leq W_1. \tag{52}$$

We use the doubling trick to expand the dimension and re-scale the samples (Step 2 in Algorithm 3). Now the original problem becomes a logistic regression problem over a probability simplex: $\Delta_{2n+1} = \{w \in \mathbb{R}^{2n+1} : \sum_{i=1}^{2n+1} w_i = 1, w_i \geq 0, \forall i \in [2n+1]\}$.

$$\min_{w \in \Delta_{2n+1}} \frac{1}{N} \sum_{i=1}^{N} -\hat{y}^i \ln(\sigma(\langle w, \hat{x}^i \rangle)) - (1 - \hat{y}^i) \ln(1 - \sigma(\langle w, \hat{x}^i \rangle)), \tag{53}$$

where $(\hat{x}^i, \hat{y}^i) \in \mathbb{R}^{2n+1} \times \{0, 1\}$. In Step 4-11 of Algorithm 3, we follow the standard simplex setup for mirror descent algorithm (see Section 5.3.3.2 of (Ben-Tal and Nemirovski, 2013)). Specifically, the negative entropy is used as the distance generating function (aka the mirror map). The projection step (Step 9) can be done by a simple $\ell_1$ normalization operation. After that, we transform the solution back to the original space (Step 12).

---

**Algorithm 3:** Mirror descent algorithm for $\ell_1$-constrained logistic regression

**Input:** $\{(x^i, y^i)\}_{i=1}^{N}$ where $x^i \in \{-1, 1\}^n$, $y^i \in \{-1, 1\}$; constraint on the $\ell_1$ norm $W_1 \in \mathbb{R}_+$; number of iterations $T$.

**Output:** $\bar{w} \in \mathbb{R}^n$.

1 **for** *sample* $i \leftarrow 1$ **to** $N$ **do**
       // Form samples $(\hat{x}^i, \hat{y}^i) \in \mathbb{R}^{2n+1} \times \{0, 1\}$.
2     $\hat{x}^i \leftarrow [x^i, -x^i, 0] \cdot W_1, \quad \hat{y}^i \leftarrow (y^i + 1)/2$
3 **end**
    // Initialize $w$ as the uniform distribution.
4 $w^1 \leftarrow [\frac{1}{2n+1}, \frac{1}{2n+1}, \cdots, \frac{1}{2n+1}] \in \mathbb{R}^{2n+1}$
5 $\gamma \leftarrow \frac{1}{2W_1} \sqrt{\frac{2 \ln(2n+1)}{T}}$              // Set the step size.
6 **for** *iteration* $t \leftarrow 1$ **to** $T$ **do**
7     $g^t \leftarrow \frac{1}{N} \sum_{i=1}^{N} (\sigma(\langle w^t, \hat{x}^i \rangle) - \hat{y}^i) \hat{x}^i$      // Compute the gradient.
8     $w_i^{t+1} \leftarrow w_i^t \exp(-\gamma g_i^t)$, for $i \in [2n+1]$      // Coordinate-wise update.
9     $w^{t+1} \leftarrow w^{t+1}/\|w^{t+1}\|_1$             // Projection step.
10 **end**
11 $\bar{w} \leftarrow \sum_{t=1}^{T} w^t / T$             // Aggregate the updates.
    // Transform $\bar{w}$ back to $\mathbb{R}^n$ and the actual scale.
12 $\bar{w} \leftarrow (\bar{w}_{1:n} - \bar{w}_{(n+1):2n}) \cdot W_1$

---

Algorithm 4 gives a mirror descent algorithm for the $\ell_{2,1}$-constrained logistic regression:

$$\min_{w \in \mathbb{R}^{n \times k}} \frac{1}{N} \sum_{i=1}^{N} \ln(1 + e^{-y^i \langle w, x^i \rangle}) \qquad \text{s.t. } \|w\|_{2,1} \leq W_{2,1}. \tag{54}$$

For simplicity, we assume that $n \geq 3$[11]. We then follow Section 5.3.3.3 of (Ben-Tal and Nemirovski, 2013) to use the following function as the mirror map $\Phi : \mathbb{R}^{n \times k} \to \mathbb{R}$:

$$\Phi(w) = \frac{e \ln(n)}{p} \|w\|_{2,p}^p, \quad p = 1 + 1/\ln(n). \tag{55}$$

The update step (Step 8) can be computed efficiently in $O(nk)$ time, see the discussion in Section 5.3.3.3 of (Ben-Tal and Nemirovski, 2013) for more details.

**Algorithm 4:** Mirror descent algorithm for $\ell_{2,1}$-constrained logistic regression

---

**Input:** $\{(x^i, y^i)\}_{i=1}^N$ where $x^i \in \{0,1\}^{n \times k}$, $y^i \in \{-1, 1\}$; constraint on the $\ell_{2,1}$ norm $W_{2,1} \in \mathbb{R}_+$; number of iterations $T$.

**Output:** $\bar{w} \in \mathbb{R}^{n \times k}$.

1 **for** *sample* $i \leftarrow 1$ **to** $N$ **do**
    // Form samples $(\hat{x}^i, \hat{y}^i) \in \mathbb{R}^{n \times k} \times \{0, 1\}$.
2     $\hat{x}^i \leftarrow x^i \cdot W_{2,1}, \quad \hat{y}^i \leftarrow (y^i + 1)/2$
3 **end**
    // Initialize $w$ as a constant matrix.
4 $w^1 \leftarrow [\frac{1}{n\sqrt{k}}, \frac{1}{n\sqrt{k}}, \cdots, \frac{1}{n\sqrt{k}}] \in \mathbb{R}^{n \times k}$
5 $\gamma \leftarrow \frac{1}{2W_{2,1}} \sqrt{\frac{e \ln(n)}{T}}$                   // Set the step size.
6 **for** *iteration* $t \leftarrow 1$ **to** $T$ **do**
7     $g^t \leftarrow \frac{1}{N} \sum_{i=1}^N (\sigma(\langle w^t, \hat{x}^i \rangle) - \hat{y}^i)\hat{x}^i$     // Compute the gradient.
8     $w^{t+1} \leftarrow \arg\min_{\|w\|_{2,1} \le 1} \Phi(w) - \langle \nabla\Phi(w^t) - \gamma g^t, w \rangle$     // $\Phi(w)$ is in (55).
9 **end**
10 $\bar{w} \leftarrow (\sum_{t=1}^T w^t/T) \cdot W_{21}$            // Aggregate the updates.

---

## K  Proof of Theorem 4 and Theorem 5

**Lemma 12.** *Let $\hat{\mathcal{L}}(w) = \frac{1}{N} \sum_{i=1}^N \ln(1 + e^{-y^i \langle w, x^i \rangle})$ be the empirical loss. Let $\hat{w}$ be a minimizer of the ERM defined in (52). The output $\bar{w}$ of Algorithm 3 satisfies*

$$\hat{\mathcal{L}}(\bar{w}) - \hat{\mathcal{L}}(\hat{w}) \le 2W_1 \sqrt{\frac{2\ln(2n+1)}{T}}. \tag{56}$$

*Similarly, let $\hat{w}$ be a minimizer of the ERM defined in (54). Then the output $\bar{w}$ of Algorithm 4 satisfies*

$$\hat{\mathcal{L}}(\bar{w}) - \hat{\mathcal{L}}(\hat{w}) \le O(1) \cdot W_{2,1} \sqrt{\frac{\ln(n)}{T}}. \tag{57}$$

Lemma 12 follows from the standard convergence result for mirror descent algorithm (see, e.g., Theorem 4.2 of (Bubeck, 2015)), and the fact that the gradient $g^t$ in Step 6 of Algorithm 3 satisfies $\|g^t\|_\infty \le 2W_1$ (reps. the gradient $g^t$ in Step 6 of Algorithm 4 satisfies $\|g^t\|_\infty \le 2W_{2,1}$). This implies that the objective function after rescaling the samples is $2W_1$-Lipschitz w.r.t. $\|\cdot\|_1$ (reps. $2W_{2,1}$-Lipschitz w.r.t. $\|\cdot\|_{2,1}$).

We are now ready to prove Theorem 4, which is restated below.

**Theorem.** *In the setup of Theorem 1, suppose that the $\ell_1$-constrained logistic regression in Algorithm 1 is optimized by the mirror descent method (Algorithm 3) given in Appendix J. Given $\rho \in (0, 1)$ and $\epsilon > 0$, if the number of mirror descent iterations satisfies $T = O(\lambda^2 \exp(12\lambda) \ln(n)/\epsilon^4$, and the number of i.i.d. samples satisfies $N = O(\lambda^2 \exp(12\lambda) \ln(n/\rho)/\epsilon^4)$, then (6) still holds with probability at least $1 - \rho$. The total run-time of Algorithm 1 is $O(TNn^2)$.*

*Proof.* We first note that in the proof of Theorem 1, we only use $\hat{w}$ in order to apply the result from Lemma 1. In the proof of Lemma 1 (given in Appendix E), there is only one place where we use the definition of $\hat{w}$: the inequality (b) in (40). As a result, if we can show that (40) still holds after replacing $\hat{w}$ by $\bar{w}$, i.e.,

$$\mathcal{L}(\bar{w}) \le \mathcal{L}(w^*) + O(\gamma), \tag{58}$$

then Lemma 1 would still hold, and so is Theorem 1.

By Lemma 12, if the number of iterations satisfies $T = O(W_1^2 \ln(n)/\gamma^2)$, then

$$\hat{\mathcal{L}}(\bar{w}) - \hat{\mathcal{L}}(\hat{w}) \le \gamma. \tag{59}$$

As a result, we have

$$\mathcal{L}(\bar{w}) \overset{(a)}{\leq} \hat{\mathcal{L}}(\bar{w}) + \gamma \overset{(b)}{\leq} \hat{\mathcal{L}}(\hat{w}) + 2\gamma \overset{(c)}{\leq} \hat{\mathcal{L}}(w^*) + 2\gamma \overset{(d)}{\leq} \mathcal{L}(w^*) + 3\gamma, \tag{60}$$

where (a) follows from (38), (b) follows from (59), (c) follows from the fact that $\hat{w}$ is the minimizer of $\hat{\mathcal{L}}(w)$, and (d) follows from (39). The number of mirror descent iterations needed for (58) to hold is $T = O(W_1^2 \ln(n)/\gamma^2)$. In the proof of Theorem 1, we need to set $\gamma = O(1)\epsilon^2 \exp(-6\lambda)$ (see the proof following (21)), so the number of mirror descent iterations needed is $T = O(\lambda^2 \exp(12\lambda) \ln(n)/\epsilon^4)$.

To analyze the runtime of Algorithm 1, note that for *each variable* in $[n]$, transforming the samples takes $O(N)$ time, solving the $\ell_1$-constrained logisitic regression via Algorithm 3 takes $O(TNn)$ time, and updating the edge weight estimate takes $O(n)$ time. Forming the graph $\hat{G}$ over $n$ nodes takes $O(n^2)$ time. The total runtime is $O(TNn^2)$. $\qquad\square$

The proof of Theorem 5 is identical to that of Theorem 4 and is omitted here. The key step is to show that (58) holds after replacing $\hat{w}$ by $\bar{w}$. This can be done by using the convergence result in Lemma 12 and applying the same logic in (60). The runtime of Algorithm 2 can be analyzed in the same way as above. The $\ell_{2,1}$-constrained logistic regression dominates the total runtime. It requires $O(TN^{\alpha,\beta}nk)$ time for each pair $(\alpha, \beta)$ and each variable in $[n]$, where $N^{\alpha,\beta}$ is the subset of samples that a given variable takes either $\alpha$ or $\beta$. Since $N \geq kN^{\alpha,\beta}$, the total runtime is $O(TNn^2k^2)$.

## L   More experimental results

We compare our algorithm (Algorithm 2) with the Sparsitron algorithm in (Klivans and Meka, 2017) on a two-dimensional 3-by-3 grid (shown in Figure 2). We experiment three alphabet sizes: $k = 2, 4, 6$. For each value of $k$, we simulate both algorithms 100 runs, and in each run we generate the $W_{ij}$ matrices with entries $\pm 0.2$. To ensure that each row (as well as each column) of $W_{ij}$ is centered (i.e., zero mean), we will randomly choose $W_{ij}$ between two options: as an example of $k = 2$, $W_{ij} = [0.2, -0.2; -0.2, 0.2]$ or $W_{ij} = [-0.2, 0.2; 0.2, -0.2]$. The external field is zero. Sampling is done via exactly computing the distribution. The Sparsitron algorithm requires two sets of samples: 1) to learn a set of candidate weights; 2) to select the best candidate. We use $\max\{200, 0.01 \cdot N\}$ samples for the second part. We plot the estimation error $\max_{ij} \|W_{ij} - \hat{W}_{ij}\|_\infty$ and the fraction of successful runs (i.e., runs that exactly recover the graph) in Figure 3. Compared to the Sparsitron algorithm, our algorithm requires fewer samples for successfully recovery.

Figure 3: Comparison of our algorithm and the Sparsitron algorithm in (Klivans and Meka, 2017) on a two-dimensional 3-by-3 grid. Top row shows the average of the estimation error $\max_{ij} \|W_{ij} - \hat{W}_{ij}\|_\infty$. Bottom row plots the faction of successful runs (i.e., runs that exactly recover the graph). Each column corresponds to an alphabet size: $k = 2, 4, 6$. Our algorithm needs fewer samples than the Sparsitron algorithm for graph recovery.

## Footnotes

[6]Lemma 5.21 in (Rigollet and Hütter, 2017) has a typo: The upper bound should depend on $\exp(2\lambda)$. Accordingly, Theorem 5.23 should depend on $\exp(4\lambda)$ rather than $\exp(3\lambda)$.

[7]This is because the Hessian of the population loss has a lower bound that depends on $\exp(-2\lambda\sqrt{k})$ for $\|w\|_{2,1} \leq \lambda\sqrt{k}$ and $\|x\|_{2,\infty} \leq 1$.

[8]For a matrix $w$, we define $\|w\|_{\infty} = \max_{ij}|w(i, j)|$. Note that this is different from the induced matrix norm.

[9]Specifically, for the $\ell_1$-constrained logisitic regression defined in (4), since the input sample satisifies $\|x\|_\infty = 1$, the loss function is $O(1)$-Lipschitz w.r.t. $\|\cdot\|_1$. Similarly, for the $\ell_{2,1}$-constrained logisitic regression defined in (11), the loss function is $O(1)$-Lipschitz w.r.t. $\|\cdot\|_{2,1}$ because the input sample satisfies $\|x\|_{2,\infty} = 1$.

[10]Other approaches include the standard projected gradient descent and the coordinate descent. Their convergence rates depend on either the smoothness or the Lipschitz constant (w.r.t. $\|\cdot\|_2$) of the objective function (Bubeck, 2015). This would lead to a total runtime of $\tilde{O}(n^3)$ for our problem setting. Another option would be the composite gradient descent method, the analysis of which relies on the restricted strong convexity of the objective function (Agarwal et al., 2010). For other variants of mirror descent algorithms, see the remark after Theorem 5.

[11] For $n \leq 2$, we need to switch to a different mirror map, see Section 5.3.3.3 of (Ben-Tal and Nemirovski, 2013) for more details.