[Reviews · NeurIPS 2019]

Reviewer 1



This paper gives a simple and elegant algorithm for solving the long-studied problem of graphical model estimation (at least, in the case of pairwise MRFs, which includes the classic Ising model). The method uses a form of constrained logistic regression, which in retrospect, feels like the "right" way to solve this problem. The algorithm simply runs this constrained logistic regression method to learn the outgoing edges attached to each node. The proof is elegant and modular: first, based on standard generalization bounds, a sufficient number of samples allows minimization of the logistic loss function. Second, this loss is related to another loss function (the sigmoid of the inner product of the parameter vector with a sample from the distribution). Finally, using a result from Klivans and Meka, they relate this function to the true error in the parameter vector. The authors also show that approximate minimization of the loss function can be done computationally efficiently via mirror descent. The sample complexity improves (slightly) upon the previous best results in case where the alphabet is k-ary (reducing the dependence on k from k^5 to k^4), but this is not the reason why I like the result. The thing I like most about this work is that it proposes and analyzes such a simple algorithm for this long-studied problem. I believe this method should be extendible and applicable to more general settings, due largely to the simplicity of its core. Aside from this, the "right" answer for such a long-studied problem definitely deserves publication. This is perhaps the main contribution -- showing that logistic regression can be used as the main primitive, rather than the more complicated Sparsitron of Klivans and Meka. The modular nature of the proof greatly helps with the clarity. This is a fairly short review, but I don't have much more to say: it's a nice result which I believe should be accepted. If I had to criticize the paper, I would say that the theoretical sample complexity is not significantly improved over previously-known methods. That said, as made clear above, I think the method itself warrants publication. It would be interesting to see a more thorough empirical evaluation, to compare with the interaction screening method and in more settings than just the very basic ones studied here. But I don't think this evaluation is necessary for acceptance.

Reviewer 2



I read your feedback. I understood that your main contribution is theoretical (and always thought it is interesting). I increased my overall score though still believe that within the eight page limit your paper could benefit from focusing on the Ising case and only briefly mention the extension to larger alphabets. But of course I know that this means a major change that is probably not feasible. This paper makes an interesting contribution. The problem formulation as l1-regularized logistic regression (in the binary case) and regularized l(2,1)-regularized softmax regression problem (in the general) is, as the authors also acknowledge, well known. The consistency (parametric and structural/sparsistency) analysis is new and improves in certain (small) aspects over previous work. Formally, the improvement over previous work is a slightly better sample complexity (the dependency on the size k of the sample spaces of the variables is only k^4 instead the previously known bound of k^5). Unfortunately, I am lacking the time to check the technical soundness of the paper. All proofs are given in an appendix that I did not read. The presentation, although the proofs are deferred to an appendix, is quite technical. The given outline of the proof is too short (at least for me) to get the underlying ideas. Experiments are only presented for rather small examples (up to 14 variables, up to k=6). Still, it is demonstrated the presented approach does indeed not need the incoherence assumption.

Reviewer 3



This work tackles the problem of learning the edge weights in general discrete pairwise MRFs, of which the classical Ising model is a particular instance (when the nodes have cardinality k = 2). A steady body of work has recently addressed this problem and made strides towards structure learning with efficient sample and time complexity. The paper contributes algorithms to learn both the structure of Ising models (using l_1 constrained logistic regression) and the general case (k > 2, using l_{2,1} constrained logistic regression). These algorithms are theoretically shown to improve upon the sample complexity of the state-of-the-art. Further theoretical results show that, when the presented mirror descent algorithm is used during optimization, the time complexity of learning may be made competitive with the state-of-the-art without sacrificing statistical guarantees. The theoretical results are supported with several experiments, which soundly show that the presented algorithms outperform the state-of-the-art in sample complexity and work with fewer model assumptions than other previous works. The paper is well written and the results contribute steady progress towards more efficiently solving this particular problem. As expected, the paper is very theoretical, but the authors did an excellent job of both keeping the main paper readable while doing further heavy lifting in the supplementary. Furthermore, the extra content in the supplementary was very helpful as a whole (e.g., Section A was very comprehensive, the extra experiments were appreciated, and the breakdown of the proofs was well done).

[Author Response · NeurIPS 2019]

We thank all reviewers for their time and valuable comments.

**Reviewer #1**

We thank this reviewer for the positive feedback!

**"The theoretical sample complexity is not significantly improved over previously-known methods."**

The main contribution of our paper is to show that an existing and popular algorithm (i.e., group-sparse regularized
logistic regression) actually gives the state-of-the-art performance (in a setting where alternative algorithms are being
proposed). We view the sample complexity improvement over the dependence on $k$ as a side benefit of our analysis.

**"It would be interesting to see a more thorough empirical evaluation, to compare with the interaction screening
method and in more settings."**

The main contribution of our paper is theoretical. A thorough empirical evaluation of different algorithms is definitely
an interesting direction for future research, and we believe is beyond the scope of our current paper. Nevertheless, we
did an experiment comparing the performance of the following algorithms: $\ell_1$-constrained logistic regression, RISE
(regularized interaction screening estimator) and its variant logRISE [LVMC18], and the Sparsitron algorithm [KM17].
Our graph has diamond shape (Figure 1 of our paper), 10 variables and edge weight 0.2. We focus on Ising models,
because RISE and logRISE *cannot* be used to learn graphical models with general alphabet. With 1500 samples, the
fraction of successful runs out of 100 runs is: 92 (logistic regression), 90 (RISE), 93 (logRISE), and 53 (Sparsitron).

**"Extend the method to higher-order MRFs."** Intuitively, it should not be difficult to prove that $\ell_1$-constrained logistic
regression can recover the structure of binary $t$-wise MRFs. One can prove it by combining results from Section 7
of [KM17] and the following fact: the Sparsitron algorithm can be viewed as an online mirror descent algorithm that
approximately solves an $\ell_1$-constrained logistic regression. This observation is actually the starting point of our paper.
For higher-order MRFs with non-binary alphabet, we conjecture that similar result can be proved for group-sparse
regularized logistic regression. Extending the current proof/method to higher-order MRFs is definitely an interesting
direction for future research. We will include this discussion in our paper.

**Reviewer #2**

**"The presentation is quite technical...the Ising case seems to be enough to introduce the main idea...but a lot of
space is devoted to the generalization to larger alphabet..."**

In this paper we consider the general alphabet setting for two reasons:

• This shows that our proof technique is actually quite general and can be easily extended to the setting with non-binary
alphabet. In fact, there is a one-to-one correspondence between the lemmas used in learning Ising models (Lemma 8,
1, 5) and the non-binary graphical models (Lemma 11, 2, 6).

• For learning non-binary graphical models, we see a benefit of using the group-sparse (i.e., the $\ell_{2,1}$-norm) constraint
instead of the $\ell_1$-norm constraint used in [KM17]: the sample complexity improves from $k^5$ to $k^4$. A more general
statement holds (by following a proof similar to ours): for any $1 \leq p \leq 2$, the $\ell_{p,1}$-constrained logistic regression
gives a $k^{3+2/p}$ dependence. The case of $p > 2$ requires a proof different from ours and it is interesting to see if one
can get a better dependence on $k$ in that case.

**"Experiments are only presented for rather small examples (up to 14 variables, up to $k = 6$)."**

The main contribution of our paper is to theoretically prove the state-of-the-art performance of an existing and popular
algorithm (i.e., group-sparse regularized logistic regression), in a setting where alternative algorithms are being proposed.
Large-scale empirical evaluation is an interesting direction, and we think is beyond the scope of our current paper.

The biggest problem with large-scale simulation is that efficiently sampling from large graphical models is difficult.
In our experiments, the samples are generated as follows: 1) We first *exactly* compute the probability distribution
defined by a graphical model with $n$ variables and alphabet size $k$; 2) We then sample from this probability distribution.
Because the distribution contains $k^n$ probabilities, the above sampling procedure is only possible for small $n$ and $k$.
When $n$ is large (e.g., $n \sim 100$), exactly computing the probability distribution is impossible, and Gibbs sampling needs
to be used. The mixing time for Gibbs sampling can be very large [BM09]. Because of this reason, we believe that
large-scale empirical evaluation of different learning algorithms is itself a contribution to this area of research.

**Reviewer #3**: We thank this reviewer for all the positive comments!

# References

[KM17] Klivans, Adam and Meka, Raghu. Learning graphical models using multiplicative weights. *FOCS*, 2017.

[BM09] Montanari, Andrea and Bento, Jose. Which graphical models are difficult to learn? *NeurIPS*, 2009.

[LVMC18] Lokhov, A. Y. et al. Optimal structure and parameter learning of Ising models. *Science advances*, 2018.


[Meta-Review · NeurIPS 2019]

A clever new approach for learning pairwise graphical models with a gain of O(k) wrt previous work (k is the alphabet size). Results seem correct (though a detailed mathematical check was not performed) and important for the problem.